



# Distinguishing the vegetation and soil component of $\delta^{13}C$ variation in speleothem records from degassing and prior calcite precipitation effects

Heather M. Stoll[1*], Chris Day[2], Franziska Lechleitner[3], Oliver Kost[1], Laura Endres[1], Jakub Sliwinski[1], Carlos Pérez-Mejías[4], Hai Cheng[4,] Denis Scholz[5]

[1]Department of Earth Sciences, ETH Zurich, Sonneggstrasse 5, 8006 Zürich, Switzerland
[2]Department of Earth Sciences, University of Oxford, South Parks Road, OX1 3AN Oxford, UK
[3]Department of Chemistry and Biochemistry & Oeschger Centre for Climate Change Research, University of Bern, Freiestrasse 3, 3012 Bern, Switzerland
[4] Institute of Global Environmental Change, Xi'an Jiaotong University, Xi'an, China
[5] Institute of Geosciences, University of Mainz, Mainz, Germany

*Correspondence to*: Heather M. Stoll (heather.stoll@erdw.ethz.ch)

**Abstract.**

The carbon isotopic signature inherited from soil/epikarst processes may be modified by degassing and prior calcite precipitation (PCP) before its imprint on speleothem calcite. Despite laboratory demonstration of PCP effects on carbon isotopes and increasingly sophisticated models of the governing processes, to date, there has been limited effort to deconvolve the dual PCP and soil/epikarst components in measured speleothem isotopic time series. In this contribution, we explore the feasibility, advantages, and disadvantages of using trace element ratios and $\delta^{44}Ca$ to remove the overprinting effect of PCP on measured $\delta^{13}C$ to infer the temporal variations in the initial $\delta^{13}C$ of dripwater. In 8 examined stalagmites, the most widely utilized PCP indicators Mg/Ca and $\delta^{44}Ca$ covary as expected. However, Sr/Ca does not show consistent relationships with $\delta^{44}Ca$ so PCP is not universally the dominant control on Sr/Ca. From $\delta^{44}Ca$ and Mg/Ca, our calculation of PCP as $f_{Ca}$, fraction of initial Ca remaining at the deposition of the stalagmite layer, yields multiple viable solutions depending on the assumed $\delta^{44}Ca$ fractionation factor and inferred variation in $D_{Mg}$. Uncertainty in the effective fractionation of $\delta^{13}C$ during degassing and precipitation contributes to uncertainty in the absolute value of estimated initial $\delta^{13}C$. Nonetheless, the trends in initial $\delta^{13}C$ are less sensitive to these uncertainties. In coeval stalagmites from the same cave spanning 94 to 82 ka interval, trends in calculated initial $\delta^{13}C$ are more similar than those in measured $\delta^{13}C$, and reveal a common positive anomaly initial $\delta^{13}C$ during a stadial cooling event. During deglaciations, the trend of greater respiration rates and higher soil $CO_2$ is captured in the calculated initial $\delta^{13}C$, despite the tendency of higher interglacial dripwater situation to favor more extensive PCP.



## 1 Introduction

In the mid- and high latitudes, changes in vegetation productivity and soil processes significantly influence the $\delta^{13}C$ of dripwater. Conditions which favor higher vegetation productivity and faster rates of heterotropic and autotrophic respiration in soils will lead to higher soil $pCO_2$ and a lower $\delta^{13}C$ of $CO_2$ compared with less productive and slower respiring systems

where the atmospheric $CO_2$ and its higher $\delta^{13}C$ will be more significant C sources in the soil. This soil signature imparted to the dripwater is also imprinted on speleothem $\delta^{13}C$. The climate sensitivity of these processes has been exploited to serve as a temperature proxy in mid-latitude speleothems(Genty et al., 2006; Genty et al., 2003). In addition to its direct effect on the $\delta^{13}C$ of soil $CO_2$, higher soil $CO_2$ concentrations also lead to more open system dissolution of karst hostrock carbonate, which further contributes to lower $\delta^{13}C$ of the dripwater DIC.

Superimposed on soil and karst process, in-cave processes subsequently modify the $\delta^{13}C$ of DIC and thereby speleothem $\delta^{13}C$ as coupled degassing and precipitation of $CaCO_3$ progressively enriches the $\delta^{13}C$ of the remaining dissolved inorganic carbon. This process has been demonstrated in lab experiments(Polag et al., 2010; Hansen et al., 2019) and can be modeled as a Rayleigh process. Extensive quantitative modeling of this processes has been conducted in purpose built models (Mühlinghaus et al., 2009, 2007; Scholz et al., 2009; Sade et al., 2022) and embedded in broader models (Owen et al., 2018). Our interest

here is to extend the previous analysis by relating the evolution of the $\delta^{13}C$ of bicarbonate to the evolution of the Ca in solution. In this way, we can evaluate the viability of using trace element and Ca isotopic data as independent indices of the progression of coupled $CO_2$ degassing and PCP.

In this contribution, we explore the potential and limitations of simple, easy to use approaches to distinguish temporal variations in $\delta^{13}C$ attributable to soil/epikarst vs in-cave processes. We acknowledge that effects to recover the absolute initial

$\delta^{13}C$ will be challenging in most applications. Therefore we focus on the simpler goal of separating the temporal trends in PCP from those due to soil/epikarst components within a given stalagmite. We present new analyses of PCP proxies $\delta^{44}Ca$ and Mg/Ca on 8 stalagmites and evaluate approaches for estimating the quantitative extent of PCP and degassing in fossil stalagmites in which initial dripwater chemistry and partitioning coefficients are not independently constrained. For three time intervals including TII, MIS5b-c, and the last deglaciation, we present $\delta^{13}C$ time series , and compare trends of coeval $\delta^{13}C$

records with and without deconvolution of in cave and soil/epikarst processes.

## 2. 2. Background on evolution of $\delta^{13}C$ with progressive $CO_2$ degassing and PCP

### 2.1 Rayleigh fractionation of carbon isotopes during $CO_2$ degassing and prior calcite precipitation

The $\delta^{13}C$ of dissolved inorganic carbon in dripwater, dominantly bicarbonate, will evolve with the coupled precipitation of calcite and evolution of $CO_2$ (g) from the dripwater. Because the C removed by $CO_2$ degassing is isotopically much lighter





than the bicarbonate pool, the $\delta^{13}C$ of the dissolved bicarbonate increases. The net total fractionation is the sum of the effective

fractionation factors ($\epsilon$) between bicarbonate and $CO_2$ (g) and between calcite and bicarbonate:

(1)  ½ $^{13}\epsilon$ $_{CO2g/HCO3-}$ + ½ $^{13}\epsilon$ $_{CaCO3(s)/HCO3-)}$

The first of these fractionation factors is strongly temperature dependent. The second, bicarbonate-calcite fractionation factor

showed negligible temperature dependence in seeded experiments (Romanek et al., 1992) but modest temperature dependence

in earlier studies and which were used in compilations (Emrich et al., 1970; Rubinson and Clayton, 1969; Mook and Rozanski,

2000).  Thus the net fractionation factor evolves from -3.5‰ at 25°C to -5.2‰ at 5°C using temperature-sensitive calcite

fractionation factors, but from -3.5‰ at 25°C to -4.6‰ at 5°C when the calcite fractionation factor is invariant.  The evolution

of the $\delta^{13}C$ of the bicarbonate in dripwater has been modeled by a Rayleigh process, in which the dependence of the $\delta^{13}C$ of

bicarbonate is a function of the remaining HCO3- in solution.  (Mühlinghaus et al., 2009, 2007).  Newer models have added

the effect of isotopic exchange between cave air and the dripwater(Hansen et al., 2017).  Finally, advection-diffusion-reaction

models simulate the dynamic speleothem formation from a flowing thin film, including the evolution from degassing,

precipitation, and exchange with atmosphere.  This latter model confirms that that the evolution of $\delta^{13}C$ of speleothem calcite

is very closely correlated to the extent of PCP once >20% of initial DIC has precipitated (Sade et al., 2022).

Previous modeling has focused on the evolution of $\delta^{13}C$ bicarbonate as a function of the remaining DIC.  In contrast, here we

focus on the evolution of $\delta^{13}C$ bicarbonate as a function of the Ca remaining in the solution, because trace element ratios and

Ca isotopic systems may conserve proxy information about the Ca remaining in solution. This effectively gives us a tracer

from which we can estimate the degree of degassing and carbonate precipitation the solution has undergone prior to deposition

of the speleothem.  CaveCalc (Owen et al., 2018) simulates this process and examples are given in Figure 1a. We therefore

relate the Rayleigh-driven progressive enrichment in $\delta^{13}C$ of bicarbonate with the progressive depletion of Ca in the dripwater.

fCa is the fraction of initial Ca remaining in solution.  This relationship is:

$$(2) \delta^{13}C_{init} = \delta^{13}C_{meas} - A \cdot ln(fCa)$$

in which $\delta^{13}C_{init}$ is the composition for a case of negligible degassing and Ca precipitation ($f_{Ca}$ =1), and $\delta^{13}C_{meas}$ corresponds

to the $\delta^{13}C$ at a given value of fCa (Table 1 provides a summary of abbreviations used in this paper). We subsequently refer to

term A as the degassing slope and explore the implications of a range of values, from the slope for equilibrium precipitation

and degassing in CaveCalc, to greater, kinetically-enhanced fractionation during degassing suggested by some

laboratory(Hansen et al., 2019) and field (Mickler et al., 2019) studies (Figure 1b; Supplemental Figure 1). The support for

each of these values and implications are discussed at length in section 5.3.



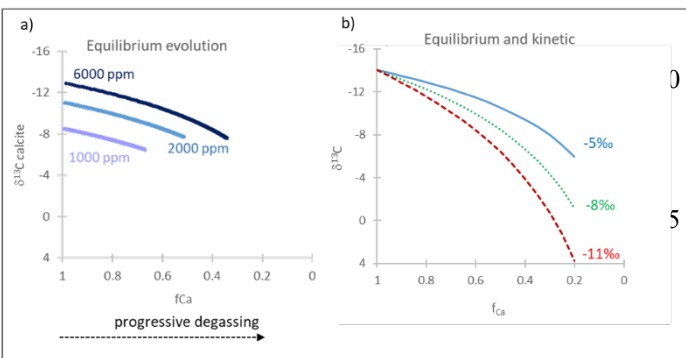

Figure 1. a) Simulation of the equilibrium evolution of $\delta^{13}C$ in calcite with increasing PCP for three initial $\delta^{13}C$ of DIC corresponding to different $pCO_2$ of soil, with the soil $CO_2$ isotopic composition following typical Keeling line (e.g. (Pataki et al., 2003). Calculations completed in CaveCalc (Owen et al., 2018). The fraction of initial Ca remaining at the time of speleothem deposition ($f_{Ca}$) is the index of PCP. b) Example simulation of the evolution of $\delta^{13}C$ in calcite from a single initial $\delta^{13}C$ of DIC following Equation (3); an equilibrium degassing and precipitation fractionation slope A (-5 ‰) is contrasted with two possible kinetically-enhanced fractionation slopes (-8 ‰ and -11 ‰).


**Table 1. List of variables and abbreviations**

| Variable | Description |
| --- | --- |
| A | degassing slope from Eqn. 2 |
| $\delta^{13}C$ meas | measured stalagmite $\delta^{13}C$ for a given sample |
| $\delta^{13}C$ init | $\delta^{13}C$ expected in stalagmite prior to degassing and PCP as calculated by Eqn. 2 |
| $\delta^{13}C\_DIC\_init$ | $\delta^{13}C$ expected in dripwater prior to degassing and PCP |
| fCa | modeled fraction of the initial Ca remaining at the time of speleothem carbonate deposition |
| fCa_$\delta$Ca | proportion of Ca remaining in solution determined using calcium isotope measurements on the stalagmite |
| fCa_MgCa | proportion of Ca remaining in solution determined using Mg/Ca measurements on the stalagmite |
| fCa (fit) | proportion of Ca remaining in solution, calculated from Mg/Ca measurements on the stalagmite with adjustments for compatibility with fCa_$\delta$Ca |
| fCa° | fCa_MgCa with AF=1 and B=1 |
| B | scaling factor to estimate bedrock Mg/Ca from minimum stalagmite measured Mg/Ca, as in Eqn. 6 |
| AF | attenuation factor to transform fCa_MgCa into fCa(fit) as in Eqn. 7 |
| Mg/Ca initial | inferred bedrock Mg/Ca and initial dripwater Mg/Ca |
| Mg/Ca min | minimum Mg/Ca measured in a given stalagmite |
| Mg/Ca meas | measured stalagmite Mg/Ca for a given sample |
| scenario A1 | scenario for which $\Delta$Ca is constant, whilst factors B and AF are adjusted to fit fCa_MgCa to fCa_dCa |
| scenario A2 | scenario for which $\Delta$Ca is constant, whilst factors B and AF are adjusted to fit fCa_MgCa to fCa_dCa |
| scenario A3 | scenario for which $\Delta$Ca is variable, whilst factors B and AF are adjusted to fit fCa_MgCa to fCa_dCa |
| scenario "full" | scenario for which AF=1 and B=1, not fit to fCa_$\delta$Ca |
| DMg | partitioning coefficient of Mg in calcite |
| DMg° | partitioning coefficient of Mg in calcite implied by fCa_MgCa with AF=1 and B=1 |
| DMg(fit) | partitioning coefficient of Mg in calcite implied by fCa(fit) as in Eqn. 8 |





## 2.2 Indicators of PCP

### 2.2.1 Mg/Ca and Sr/Ca

As discussed previously (Stoll et al., 2012), the signal of PCP imprinted on a stalagmite includes calcite precipitation truly prior to impingement of the water drop on the stalagmite surface, as well as the "extent of precipitation" occurring from the drop on the stalagmite surface before it is displaced from the active growth axis by flow. As the combined signal is manifest in the stalagmite, hereafter we discuss both true PCP and the extent of precipitation processes under the term "PCP".

Mg/Ca is the most widely applied indicator of degassing and PCP. Due to the low partitioning coefficient of Mg in calcite (e.g. as low as 0.012, (Day and Henderson, 2013)), the precipitation of calcite leads to increase in the solution Mg/Ca. The Rayleigh equation (2) relates the fraction of initial Ca remaining in solution (fCa_MgCa) to the initial dripwater Mg/Ca (eg prior to degassing) and measured solution Mg/Ca , when the partitioning coefficient D is known.

$$(3) \ f_{Ca\_MgCa} = \frac{Mg/Ca_{speleothem}}{Mg/Ca_{initial} \cdot D_{Mg}}^{\frac{1}{(D_{Mg}-1)}}$$

Figure 2a illustrates an example evolution. Mg/Ca is most robust as a quantitative indicator of PCP in a given stalagmite when the initial dripwater Mg/Ca (prior to degassing) remains constant, and when there is minimal variation in the partitioning coefficient of Mg. In cave analogue laboratory experiments, the Mg partitioning coefficient increases by about 17% with a 10°C temperature increase (Day and Henderson, 2013). Additionally, in farmed calcite a 60-70% increase in the DMg was observed as calcite Mg/Ca increases from 10 to 30 mmol/mol (Wassenburg et al., 2020). Such a dependence would serve to 120 amplify the Mg/Ca due to increasing PCP. This dependence of DMg on Mg/Ca has also been shown in some non-cave analogue laboratory experiments (Alkhatib et al., 2022). For inference of PCP, the measured Mg in the stalagmite should be fully in the calcite and not in detrital minerals, a criteria which can be effectively checked using other detrital sensitive indicators like Al/Ca.

A similar formulation to Eq (1) can be made for other divalent cations with partitioning coefficients significantly less than 1, 125 such as Sr/Ca and Ba/Ca. The coherence of changes in Mg/Ca, Sr/Ca and to a lesser extent Ba/Ca is often interpreted to signify that PCP is the controlling process(Sinclair, 2011; Wassenburg et al., 2020). Deviations from expected PCP control may reflect variation in the partitioning coefficients of Sr (or Ba). Increased DSr with higher growth rate or saturation state is widely seen in non-cave analogue laboratory experiments (Tang et al., 2008a; Tesoriero and Pankow, 1996; Lorens, 1981). An up to 5-fold increase in DSr is simulated for a two order of magnitude increase in growth rate (Nielsen et al., 2013) based on ion by 130 ion growth models and non-cave analogue experiments. According to this model, the magnitude of the growth rate dependence depends on the forward and reverse reaction rates which are sensitive to temperature and solution chemistry. Additionally, in-cave measurements suggest that the DSr increases with increasing calcite Mg/Ca ratio. A tripling of calcite Mg/Ca from 1.4





to 4.2mmol/mol leads to an 18% increase in DSr, and a tripling of MgCa from 10 to 30 mmol/mol leads to an 80% increase in DSr (Wassenburg et al., 2020).

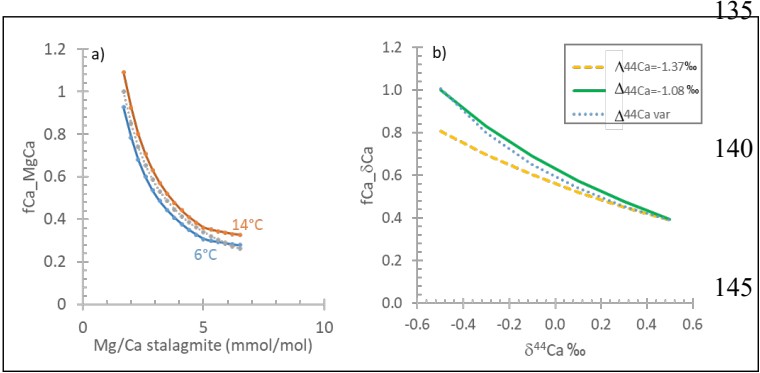

**Figure 2. a) . Example variation in Mg/Ca with fCa from Rayleigh simulation of progressive calcite precipitation at 6°C (lowermost curve) and 14°C (uppermost curve) using DMg of (Day and Henderson, 2013); example is shown for Mg/Ca dripwater 400 mmol/mol. Also shown with dashed line is the "index" approximation produced by equation 5. b) Example estimation of fCa from δ44Ca, illustrating effect of fractionation factor on the calculated fCa for two examples of constant fractionation, and one case in which the fractionation factor systematically evolves from Δcalcite-dissolved. of -1.37 to -1.08 with increasing PCP and concomitant slowing of growth rate. Example shown for a bedrock δ44Ca of 0.58.**

### 2.2.2 Calcium isotopes

$\delta^{44}Ca$ has also been employed as an indicator of PCP, since there is a fractionation of Ca during the dehydration of the ion for it to be incorporated in the crystal lattice. The lighter isotope desolvates more rapidly and thereby is incorporated preferentially in the calcite lattice. Consequently, with progressive removal of Ca into calcite, the remaining dissolved Ca evolves to isotopically heavier composition(Owen et al., 2016) (Figure 2b). The quantitative calculation of fCa from calcium isotopes (fCa_$\delta$Ca) is most robust when the Ca isotopic composition of the host rock is known and initial $\delta^{44}Ca$ from dissolution is constant (ie always congruent), and when the fractionation factor for Ca isotopes during dehydration and calcite precipitation is constant and known. The fractionation factor may be expressed as $\alpha$, or as $\Delta_{calcite-dissolved}$.

Previous applications of $\delta^{44}Ca$ in speleothem studies have focused on the Holocene, and have monitored cave dripwater and farmed calcite to calculate the dissolved-calcite fractionation factor in that particular cave setting and drip site(Owen et al., 2016; De Wet et al., 2021). Studies then apply this constant fractionation factor to calculate fCa_$\delta$Ca in stalagmite. For fossil, non-active stalagmites, the fractionation factor cannot be measured in farmed calcite and must be assumed. A summary of field and experimental fractionation factors is given in Supplementary Table 1. In one monitoring study, $\Delta_{calcite-dissolved}$. of farmed calcite varied among different drips in a modern cave(De Wet et al., 2021). Models of desolvation and attachment during crystal growth suggest that at higher net precipitation rates, the effective fractionation is greater (Depaolo, 2011). This has been observed in non-cave analogue laboratory experiments (Tang et al., 2008b), and simulated in models which suggest that a two order of magnitude increase in net precipitation rates lead to a decrease in $\Delta_{calcite-dissolved}$ from -0.75 to -1.45 ‰(Mills et al., 2021). According to this model, the magnitude of the growth rate dependence – specifically the absolute range of growth rate at which $\Delta_{calcite-dissolved}$ varies strongly - depends on the forward and reverse reaction rates which are sensitive to temperature and solution chemistry(Depaolo, 2011). For a given growth rate, lower temperatures strengthen the water

solvation structure, leading to greater fractionation. Laboratory study suggests that a 10 degree cooling leads to a $\Delta_{44-40Ca}$

decrease of 0.17‰ (Tang et al., 2008b). Yet over a seasonal 7.5C cave air temperature cycle in Heshang Cave, no seasonal

difference in $\Delta_{44-40Ca}$ was resolved within analytical error (Owen et al., 2016).

**2.2.3 Previous comparisons of PCP indicators**

Both $\delta^{44}Ca$ and Mg/Ca have been simultaneously employed to study Holocene stalagmites (HS4 in Heshang Cave, growing

about 170 microns/year(Owen et al., 2016) and last interglacial stalagmites from Northern India (Magiera et al., 2019). In

these two settings, there is good quantitative agreement in estimated fCa_δCa and fCa_MgCa. In these settings, in which

temperature and dripwater initial oversaturation may not have experienced significant variations, calcite-water Δ44Ca

fractionation factors may have remained constant. Likewise, although Heshang Cave contains dolomite host rock, variations

in water/rock contact times did not result in appreciable variation in the initial Mg/Ca from dissolution (ie not significant

enough to cause deviation from theoretical expectation).

**2.3 Limitations of the Hendy Test for degassing correction**

The sampling of calcite precipitated from the same initial dripwater but with progressive degree of $CO_2$ degassing and calcite

removal has often been attempted through extraction of samples in a single growth layer at succesive distances along the

growth axis (Hendy, 1971). In theory, the evolution of Mg/Ca and $\delta^{13}C$ could be examined along a growth layer. However, in

practice it is difficult and often impossible to follow a single growth layer laterally along the stalagmite, especially since growth

layer thickness decreases with increasing distance from the main growth axis(Dorale and Liu, 2009). Furthermore, simulations

suggest that for a drip interval of 60s, a flow distance in excess of 3 cm from the main axis is required to generate even a 1

permil enrichment of $\delta^{13}C$ through PCP and $CO_2$ degassing (Mühlinghaus et al., 2009). Finally, because the dripwaters

experiencing a lot of degassing are less saturated as they flow off axis than drips with little degassing, the off-axis growth may

be biased towards times of low degassing and low PCP. For these reasons, it is often impractical to test relationships between

$\delta^{13}C$ and Mg/Ca using the Hendy test.

**3. Methods**

**3.1 Geochemical models of speleothems**

We employ CaveCalc for simulations of equilibrium fractionation of carbon isotopes. The CaveCalc package employs the

PHREEQC geochemical model to simulate the initial dissolution of karst limestone in equilibrium with a given volume of soil

gas of specified $pCO_2$ and $\delta^{13}C$, and the subsequent equilibration of this solution with a cave atmosphere of specified $pCO_2$

leading to precipitation of $CaCO_3$. Relevant for this study, the CaveCalc package calculates the initial $\delta^{13}C$ of bicarbonate

and initial Ca concentration of the dripwater resulting from karst dissolution, and the stepwise evolution of both parameters as



well as the $\delta^{13}C$ of the precipitated calcite as the solution equilibrates with the cave atmosphere. These CaveCalc simulations
allow us to relate the Rayleigh-driven progressive enrichment in $\delta^{13}C$ of bicarbonate with the progressive depletion of Ca in
the dripwater in equilibrium conditions.   CaveCalc employs the temperature dependent calcite- $HCO_3^-$ fractionation as
calculated by (Mook and Rozanski, 2000).

Independent from CaveCalc, a modified version of the I-STAL model (Stoll et al., 2012) is used to simulate PCP variation
resulting from changes in initial dripwater saturation state, and changes in drip interval.   The effect of variations in initial
dripwater Ca, temperature, and drip interval are explored at conditions of cave $pCO_2$ fixed at ventilated, near interglacial
atmospheric concentrations (300 ppm).   In I-STAL calculations, the temperature sensitivity of DMg follows (Day and
Henderson, 2013).

### 3.2 Analysis of fossil stalagmites

We compile existing and report new data on 9 stalagmites from NW Spain with growth periods during the Holocene, during
late MIS 5, and during the penultimate deglaciation (Table 2). Reported stalagmites are from La Vallina Cave, whose setting
and lithology are described in a detailed previous monitoring study (Kost et al). The cave is hosted in Carboniferous limestones
of the Barcaliente Formation.  Because the section is dipping at 80°, different sectors of the cave sample stratigraphically
different portions of the limestone, which have significant heterogeneity in Mg/Ca and some heterogeneity in Sr/Ca and other
trace elements.  Thus, different drip locations can feature different initial trace element ratios due to congruent dissolution of
limestones of differing composition.  Despite site to site heterogeneity, individual drips monitored over an 18 month period
show very limited temporal variation in Mg/Sr ratios despite order of magnitude differences in drip rate, suggesting that in this
cave, individual drips sample a relatively stable bedrock dissolution source.  For most stalagmites, age models are published
previously, including those for GAE, GAL, and GLO (Stoll et al., 2015; Stoll et al., 2013), GAR and GUL (Stoll et al., 2022).
For BEL, GLD, and ROW, age model constraints are given in the Supplementary Table 2.
For geochemical analyses, we sampled approximately 1 mg of powder. Trace element ratios (Mg/Ca, Sr/Ca) were determined
by dissolution of powders in 2% HNO3 and analysis on Agilent 8800 ICP-MS at ETH Zurich in collision mode. On splits of
the same powders, we measured $\delta^{13}C$  and $\delta^{18}O$ using a Thermo Fisher Scientific Gas Bench II with methods previously
described(Breitenbach and Bernasconi, 2011).

In 8 stalagmites, we selected several intervals of constrasting Mg/Ca to additionally measure $\delta^{44}Ca$.  From the same aliquot
used for Mg/Ca analysis, ~500 μg of powder was dissolved in distilled 2M $HNO_3$. An automated Ca-Sr separation method
(PrepFAST MC, Elemental Scientific, Omaha, NE, USA) as used to separate Ca from Sr, Mg and other matrix elements, to
avoid isobaric interferences during multi-collector inductively coupled mass spectrometry (MC-ICP-MS). SRM 915b solutions
were purified in parallel with the samples to provide a combined column-chemistry and analytical accuracy assessment. Ca-
isotope ratios were determined using a Nu Instruments MC-ICP-MS (the University of Oxford) with a desolvating nebulizer
as described previously. (Reynard et al., 2011).  Solutions were run at 10 ± 1 ppm concentration, and the samples were





measured with standard-sample bracketing. A minimum of 5 analyses were conducted on each sample. δ44/40Ca is reported normalized to NIST SRM 915a and was calculated from measured δ$^{44/42}$Ca, as δ$^{44/40}$Ca = δ$^{44/42}$Ca * ((43.956-39.963)/(43.956-41.959))(Hippler et al., 2003). To determine accuracy and external precision, secondary standards NIST SRM 915b and HPSnew (in-house standard) were used. Uncertainty on Ca isotope data is quoted as the t-distribution-derived 95% confidence

interval on the mean of repeat measurements calculated using either the standard deviation on all repeat measurements on each sample.

Additionally, we incorporate published δ$^{13}$C, fCa_δCa, and Mg/Ca data from Heshang stalagmite HS4(Owen et al., 2016; Noronha et al., 2014).

### 3.4 Calculation of fCa from δ$^{44}$Ca, Mg/Ca and Sr/Ca

Mg/Ca is the preferred measurement for deriving a continuous record of δ$^{13}$C$_{init}$ because measurement is much faster and therefore it is common to have Mg/Ca measured for every δ$^{13}$C$_{init}$ measured in the stalagmite. At the same time, using δ$^{44}$Ca to derive the PCP-corected δ$^{13}$C$_{init}$ may be more robust, because the δ$^{44}$Ca of the initial solution can be derived from the measured bedrock and the isotopic composition of the bedrock is expected to be more homogeneous than Mg/Ca leading to less uncertainty in the initial solution, and because Ca is a major element.

We calculate fCa_δCa from δ$^{44}$Ca measurements using a constant bedrock δ$^{44}$Ca equal to average bedrock (0.58‰) reported previously (Lechleitner et al., 2021b). Use of the highest or lowest measured bedrock δ$^{44}$Ca leads to a +/- 0.05 range in absolute fCa_δCa values. Because bedrock δ$^{44}$Ca is not expected to change at a given location over time, we do not expect this factor to contribute to temporal variations in δ$^{44}$Ca. The Δ$_{calcite-dissolved}$. for these fossil stalagmites is not independently constrained. Therefore, we complete a sensitivity analysis using the range of Δ$_{calcite-dissolved}$. observed in laboratory calcite

growth and ion by ion growth models. Supplementary Table 1 lists the 4 values of Δ$_{calcite-dissolved}$. which we evaluate.

Inference of the fCa_MgCa from Mg/Ca using the Rayleigh formula requires knowledge of the partitioning coefficient and the initial dripwater Mg/Ca, which are both difficult to infer for fossil stalagmites.

Because the DMg is very low, and the exponent in the Rayleigh formula is therefore very close to -1, the following provides a close approximation for fCa: :

(4)    $f_{Ca\_MgCa} = \frac{MgCa_{initial}}{MgCa_{sample}}$

For a DMg of 0.025, this approximation deviates from the Rayleigh equation by 0.01 at fCa =0.36 and deviates by lesser degrees for higher fCa_MgCa (e.g. gray dashed line in Figure 2a).

Because of heterogeneity in cave bedrock, we calculate fCa_MgCa assuming there may be differences in the initial dripwater Mg/Ca for each stalagmite location. Time series data for each stalagmite indicates a range of Mg/Ca values. In a first approach,

we assume that the minimum Mg/Ca of a stalagmite corresponds to a situation of negligible degassing and PCP, and that the DMg is constant, and therefore that the numerator can be approximated by the minimum Mg/Ca for the stalagmite:



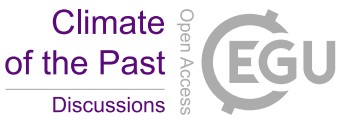

(5) $\qquad f_{Ca\_MgCa} = \dfrac{MgCa_{min}}{MgCa_{sample}}$

By default in this approximation, the maximum fCa_MgCa calculate for any stalagmite is 1. Therefore, we consider the consequences for calculated fCa_MgCa if the measured minimum Mg/Ca in the stalagmite corresponds to precipitation from

a solution which has already experienced PCP. We assess this impact with a scaling factor B (between 0.5 and 1) to approximate the initial bedrock dissolution Mg/Ca from the minimum Mg/Ca measured in the stalagmite, and apply a constant B for the stalagmite:

(6) $\quad MgCa_{initial} = MgCa_{min} * B$

fCa_SrCa may also calculated from Sr/Ca measurements in an approach analogous to Mg/Ca. Because of the higher DSr,

Equation 5 is a less accurate approximation for Sr, eg for a DSr of 0.1, there would be a 0.04 deviation of the fCa_SrCa by Eq. 5 (fCa =0.30) compared to the Rayleigh formula at fCa_SrCa=0.265

Finally, we also evaluate the possibility of temporal variation in DMg in a given stalagmite, testing the scale of variation in DMg which would be consistent with fCa_$\delta$Ca estimations from $\delta^{44}$Ca. In this fCa(fit), the curvature or slope of the relationship in Figure 2a is modified by an attenuation factor AF.


$$(7) \qquad f_{Ca\,(fit)} = B - \left( \dfrac{\left( B - \dfrac{MgCa_{initial}}{MgCa_{sample}} \right)}{AF} \right)$$

The relative change in DMg needed for the fit is given as:

$$(8) \qquad \dfrac{D_{Mg(fit)}}{D_{Mg°}} = \dfrac{f_{Ca(fit)} MgCa_{init°}}{f_{Ca°} MgCa_{init(fit)}}$$

Where fCa° indicates the fCa_MgCa for the full scenario, with AF=1 and B=1. This AF gives a linear increase in DMg with

Mg/Ca, however other forms of dependence could also fit the data, since we have $\delta^{44}$Ca for only 2 points for most of the analyzed stalagmites.

### 3.5 Estimation of $\delta^{13}C_{init}$ from $\delta^{13}C_{meas}$ and estimated fCa

We use Mg/Ca to derive continuous PCP-corrected $\delta^{13}C_{init,}$ but we employ the few existing $\delta^{44}$Ca measurements for each stalagmite to validate and adjust the precise relationship between Mg/Ca and PCP yielding fCa (fit) in order to achieve a more

robust estimate of $\delta^{13}C_{init.}$ In detail, we use the Mg/Ca to calculate fCa in 4 different ways: fCa_MgCa from the minimum stalagmite Mg/Ca and constant DMg (termed "full" with both B and AF=1), and additionally from three other fCa(fit) using equation 7 (termed "A1, A2, and A3") in which the parameters B and AF are adjusted to provide fCa_MgCa compatible with fCa_$\delta$Ca. For each stalagmite, the values of B and AF employed for A1, A2, and A3 are given in Table 3.





For each of these four possible fCa scenarios, we additionally calculate $\delta^{13}C_{init}$ using three possible values of the degassing slope (A) in equation 3: -5, -8, and -11 ‰ (e.g. in Figure 2b). This exercise illustrates the consequences of a range of possible equilibrium and disequilibrium fractionation behaviors.

## 4. Results

### 4.1 PCP indicators in stalagmites

Within the subsamples measured for both Mg/Ca and $\delta^{44}Ca$, the $\delta^{44}Ca$ of measured samples ranges from -0.49 to +0.42. In all of these subsets, the Mg/Ca and $\delta^{44}Ca$ positively correlate (Figure 3). In all but two stalagmites, Sr/Ca and $\delta^{44}Ca$ positively correlate. In GUL and GAL, Sr/Ca and $\delta^{44}Ca$ negatively correlate.

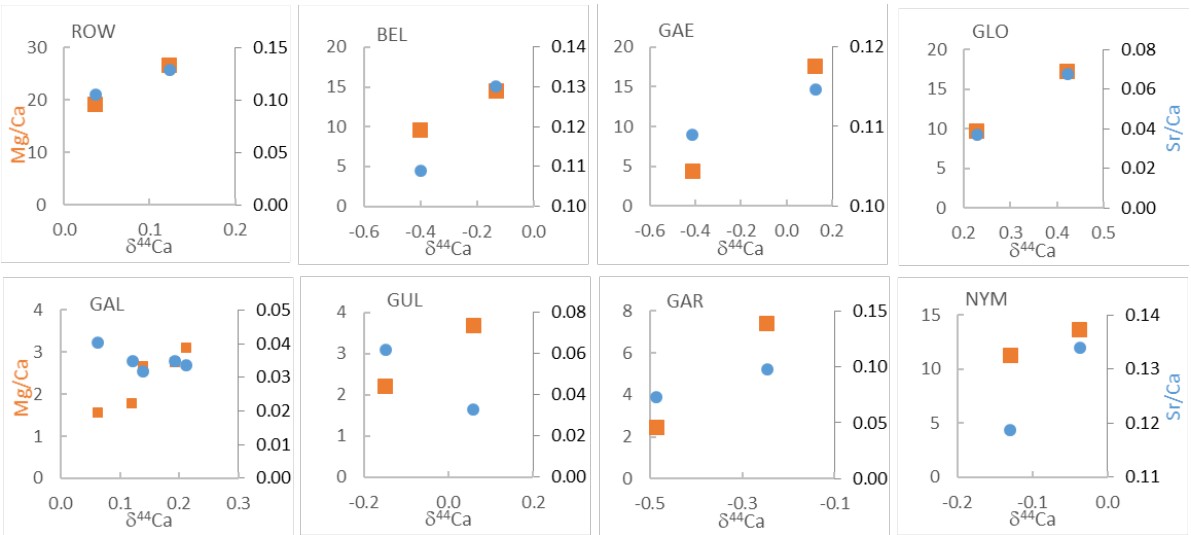

**Figure 3. Mg/Ca (orange squares on left axes, in mmol/mol) and Sr/Ca (blue circles on right axes, mmol/mol) vs $\delta^{44}Ca$ (‰) for paired samples from each stalagmite.**

Across the full geochemical sampling of the 8 speleothems from Porrua Cave, the total range in Mg/Ca across each speleothem is between 1.7 and 3.5 in all speleothems except GAE (12.2); 5 of the stalagmites have a range between 2 and 3 (Table 2). The 1.7 to 3.5 fold range could be fully explained by PCP even with no change in DMg, since a 3.5 fold range requires the initial dripwater Ca concentration to be greater than 3.5 times the saturation Ca concentration for the cave conditions. Changes in the Mg/Ca of the initial dripwater, prior to degassing, from process such as enhanced water rock interaction, or enhanced Mg/Ca ratio due to increased fluid inclusion density, are not required to attain the range of Mg/Ca in 7 of the 8 of the stalagmites. With the exception of Holocene GAL, minimum Mg/Ca generally occurs during the glacials and stadials (Table 2). In GAE, the basal 0.5 cm of the stalagmite feature an unusually low Mg/Ca ratio which is 12-fold lower than the maximum ratio. If driven solely by PCP, this range would require dripwater Ca in excess of 300 ppm during warm periods to drive such a large range in PCP. In this stalagmite, the Mg/Ca of this basal portion may reflect a different initial state of the fracture





network and mineral surface age over time. GAE was sampled growing on a large block fallen from the ceiling and first growth
        after block emplacement may reflect flow through newly opened fracture networks.

        The total range in Sr/Ca across each speleothem is between 1.7 and 4 in all speleothems except GAE (6) (Table 2).  The
        strongest positive correlation between Sr/Ca and Mg/Ca is found in GLO, GLD, and BEL.  Strong negative correlation occurs
        in GUL, and modest negative correlation occurs in GAL and GAR.


**Table 2: Sample information and trace element summary.**

| stalagmite | period included in this study | age base and tip | min Mg/Ca (mmol/mol)* | age min Mg/Ca (ka) | max Mg/Ca (mmol/mol) | age max Mg/Ca (ka) | range Mg/Ca (max/min) | min Sr/Ca (mmol/mol) | max Sr/Ca (mmol/mol) | range Sr/Ca (max/min) | Correlation Sr/Ca and Mg/Ca** |
|---|---|---|---|---|---|---|---|---|---|---|---|
| GAL | 9-4 ka, 26 ka | 26 ka, 1 ka | 1.1 | 2 | 3.8 | 6 | 3.5 | 0.023 | 0.054 | 2.3 | -0.29 |
| GUL | 14.5 - 4 ka | 14.5 ka to 4 ka | 1.6 | 14 | 4.1 | 5 | 2.5 | 0.026 | 0.065 | 2.5 | -0.71 |
| GAE | 94-82 ka | 135 ka to 73 ka | 2.2 | 143 | 26.3 | 129, 97 | 12.2 | 0.025 | 0.162 | 6.5 | 0.37 |
| GAR | 140-125ka | 217 ka to 112 ka | 2.8 | 145 | 8.5 | 127 | 3.0 | 0.028 | 0.094 | 3.3 | -0.34 |
| GLO | 94-82 ka | 196 ka to 84 ka | 7.6 | 166 | 22.5 | 85 | 3.0 | 0.020 | 0.081 | 4.0 | 0.68 |
| BEL | 140-125ka | 172 ka to 128 ka | 9.1 | 134 | 25.3 | 131 | 2.8 | 0.076 | 0.247 | 3.3 | 0.72 |
| NYM | MIS 5e | 148 ka to 113 ka | 9.3 | 115 | 18.6 | 122 | 2.0 | 0.088 | 0.162 | 1.8 | 0.45 |
| ROW | 94-82 ka | 107 ka to 82 ka | 18.1 | 108 | 30.4 | 81 | 1.7 | 0.095 | 0.156 | 1.6 | -0.1 |
| GLD | 140-125ka | 141 ka to 99 ka | 18.8 | 136 | 40.0 | 96 | 2.1 | 0.066 | 0.153 | 2.3 | 0.77 |
| *including all analyses for the given stalagmite | | | | | | | | | | | |
| ** over the period of interest | | | | | | | | | | | |

## 4.4  Calculated fCa from PCP indicators

### 4.4.1. Expected ranges in fCa

        Because highly oversaturated dripwaters have a greater potential for PCP than minimally oversaturated dripwaters, fCa can
        vary over a wider range in settings with high oversaturation (warm climates with higher soilCO$_2$ and initial dripwater Ca;
        Figure 4a). In contrast, minimum degassing and high fCa will be favored by very low oversaturation state of the drip, even for
        slow drip rates (Figure 4a, b).  Minimal degassing is also favored by colder temperatures.  In the mid and high latitudes, low
oversaturation of dripwaters and low PCP are more likely during cold glacial or stadial time periods when CO$_2$ in soils is
        depressed due to low temperatures (e.g Fig. 4b).  We consider this expected association of colder climates and higher fCa in
        the quantitative interpretation of fCa indices.  Alternatively, when soil pCO$_2$ and dripwater initial saturation are regulated by
        the moisture limitation of soils, then fCa varies over a narrower range at constant drip interval (Fig. 3b).  We simulate fCa as
        low as 0.15 in the case of an initial Ca concentration of 137 ppm.



Because the relationship between drip interval and PCP is known to be highly nonlinear (Fig 4c, d), different coeval stalagmites often have asynchronous variations in PCP indicators or contrasting magnitudes of PCP variation.

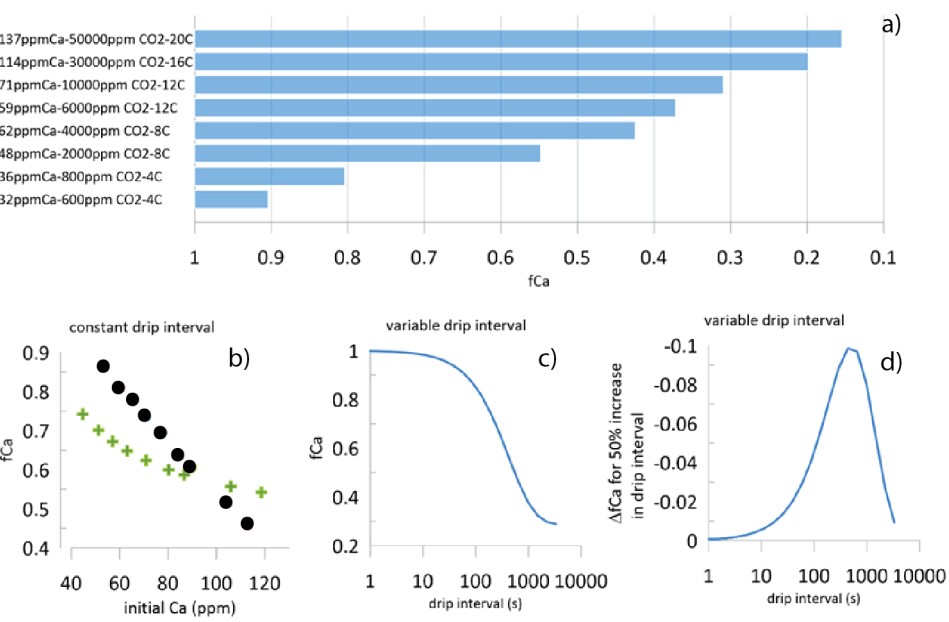

**Figure 4. a) The degree of PCP which is possible based on the initial dripwater saturation state simulated by CaveCalc for the initial ppm Ca, soil pCO2 for 150 L gas volume, and reaction temperature. The lowest fCa is defined as that which would correspond to**
**an instantaneous calcite deposition rate equivalent to 4 μm/year, as simulated in I-STAL with Dreybrodt(Romanov et al., 2008) kinetics. b) the fCa simulated for a fixed drip interval of 300 s, but variable initial saturation (indexed by Ca concentration). In one case shown with circles, the variable initial Ca corresponds with a progressive decline in temperature from 18°C to 5°C simulating soil pCO₂ limited by temperature. In a second case shown with green crosses, variable initial Ca corresponds to constant temperature, simulating soil pCO₂ limited by moisture at constant temperature. Simulations assume Dreybrodt kinetics (Romanov**
**et al., 2008)executed in I-STAL with PCP enhancement factor of 3. c, d) Illustration of nonlinearity of PCP relative to drip interval, simulated with Dreybrodt kinetics at temperature 12C, initial Ca 90 ppm, d=0.01 and PCP enhancement factor of 3. The drip interval range of maximum PCP sensitivity will vary with modeled temperature and PCP enhancement parameters.**

### 4.4.1 $\delta^{44}$Ca

The absolute fCa_$\delta$Ca calculated from a given $\delta^{44}$Ca depends on the choice of calcite-dissolved fractionation factor
(Supplemental Table 1). Given bedrock estimated at 0.58 ‰, the measured $\delta^{44}$Ca in some samples would imply an fCa_$\delta$Ca higher than one for the $\Delta$44Ca fractionation factor corresponding to slowest laboratory growth rates (-0.66.). For a given stalagmite, choice of a $\Delta$44Ca corresponding to slower laboratory growth rates (-0.66, -0.86) yields a wider range of calculated fCa_$\delta$Ca than choice of $\Delta$44Ca corresponding to faster laboratory growth rates (-1.08, -1.37). The largest range in fCa_$\delta$Ca is found in GAE, and for a given $\Delta$44Ca, the lowest average fCa_$\delta$Ca are in GAL and GLO (Figure 5, Appendix A). Because
the fractionation factor depends on growth rate and the solution characteristics, it may not be constant throughout the growth period of a stalagmite if there are significant changes in the growth conditions. For a given stalagmite, slower growth rate during periods of low initial dripwater oversaturation might be characterized by less fractionation e.g. a $\Delta$44Ca which is closer to 0, compared to periods of stalagmite growth from solutions with greater oversaturation.





### 4.4.2 Mg/Ca and Sr/Ca

Using Equation 4 and the minimum Mg/Ca for each stalagmite as an estimation of a nondegassed, fCa_MgCa=1, the high

Mg/Ca stalagmites BEL and ROW exhibit a range in fCa_MgCa from 0.95 to 0.6 (Figure 5; Appendix A) and GLO exhibits a

similar range (Appendix A). As commented previously, in GAE, the total Mg/Ca range exceeds that expected from PCP if the

minimum Mg/Ca in the basal growth phase is used. We complete sensitivity analysis calculating fCa_MgCa of GAE with the

$5^{th}$ percentile value rather than minimum, which is equivalent to the minimum Mg/Ca observed in the upper 82 cm of the

stalagmite, including the period of interest presented here. Garth exhibits the highest range in fCa_MgCa estimated from

measured Mg/Ca. A smaller range of fCa_MgCa and lowest fCa_MgCa is calculated from Equation 6 when the B factor is

<1.

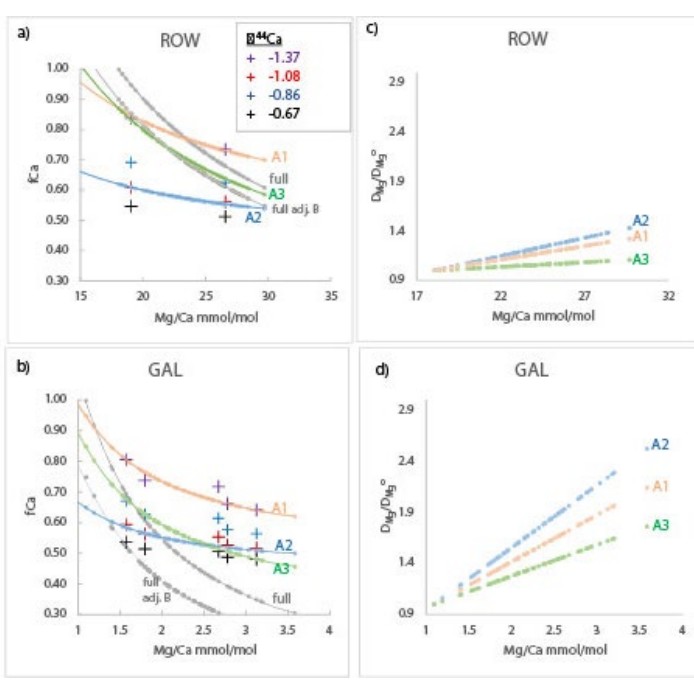

**Figure 5. a)-b) fCa_δCa (crosses) and fCa_MgCa and fCa(fit) (both with lines) vs measured Mg/Ca. Crosses show fCa_δCa calculated from $\delta^{44}$Ca according to different $\Delta^{44}$Ca fractionation factor, vertically ordered according to fractionation factor as given in the legend. Gray curves show calculation of fCa_MgCa from Mg/Ca assuming Mg/Ca$_{min}$ reflects undegassed dripwater ("full", upper gray line) or that undegassed dripwater is lower than minimum Mg/Ca by factor from 0.65 to 0.95 ("full adj. b", lower gray line; value in Supplemental Table 1). Pink, blue, and green lines illustrate scenarios A1, A2, and A3 which are potential relationships between fCa(fit) and Mg/Ca consistent with fCa_δCa estimates according to Equation 7 and fit parameters in Table 3. c)-d) The variation in DMg implied by scenarios in a)-b),calculated as in Equation 8 and assuming constant congruent bedrock dissolution to yield a constant initial undegassed Mg/Ca ratio of dripwater. A 10°C temperature increase would cause D/D$_0$ to reach 1.22 according to laboratory experiments(Day and Henderson, 2013).**


### 4.4.3 Comparison of fCa estimations

We explored which combinations of $\Delta 44$Ca and assumptions of B and AF lead to coherent estimations of fCa_δCa and

fCa_MgCa of yield fCa(fit). We find that in many stalagmites, with AF=1 the Mg/Ca leads to a wider range in fCa_MgCa

than fCa_δCa (e.g. Figure 5, supplemental Figure 1). Covariation in DMg with Mg/Ca is one process which could cause AF

to be greater than 1.

Estimates of fCa_δCa and fCa_MgCa for stalagmites BEL, ROW, and NYM are consistent with no systematic variation in

DMg (ie an AF ≈1) if the calcite-dissolved fractionation factor $\Delta 44$Ca varies by 0.2‰ between the sample of lowest and





highest PCP (Scenario A3; Figure 5, Table 3). Alternatively, if the Δ44Ca were constant (Scenario A1, A2), then fCa_δCa and fCa_MgCa estimates are consistent only when AF is 1.25 to 3, which imply a systematic variation in DMg with Mg/Ca
(Figure 5). For other stalagmites (Supplemental Figure 1), if the calcite-dissolved fractionation factor Δ44Ca varies by 0.2‰ between the sample of lowest and highest PCP (Fit A3), some variation in DMg is still required (AF 1.5-2.2); for assumption of constant Δ44Ca in each stalagmite, a larger range of AF is required (1.5 to 3, with a single higher value of 4 required for GAR). For GAE, only choice of a Mg/Camin of 6 mmol/mol, the 5th percentile value, enabled calculation of PCP factor consistent with $\delta^{44}$Ca.

The concomitant changes in DMg for these scenarios ranges from 1 (eg constant DMg) to over 2.5-fold increase in DMg over the range of Mg/Ca in the stalagmite (Figure 5). GAL and GAR have the most significant increases in DMg, wherease BEL, NYM, and ROW feature the lowest. Most scenarios require an increase in DMg larger than that expected from reasonable (5 to 10 degree) temperature dependence of DMg alone; for example according to experimental cave-analogue calculation, a 10 degree warming would lead to a $D_{Mg}/D_{Mg}o$ of 1.22(Day and Henderson, 2013). Holocene stalagmite GAL features large
simulated range in DMg despite only limited regional temperature change.

We acknowledge that we have limited fCa_δCa for each stalagmite, and therefore our particular scenarios are intended to illustrate potential compatible solutions but do not cover all the possible ranges. The variation in DMg may be exaggerated because datasets with higher numbers of paired $\delta^{44}$Ca and Mg/Ca such as GAL show coherency but also some scatter around this relationship, which may be interpreted as a steeper or shallower relationship in our limited dataset.

**Table 3. Parameters for elaboration of fCa based on $\delta^{44}$Ca and Mg/Ca**

| | full | B1 | | Scenarios | | | | | | | | |
| | | | | A1 | | | A2 | | | A3 | | | |
| | B | B | Δ44Ca | B | AF | Δ44Ca | B | AF | Δ44Ca (1) | D44Ca (2) | B | AF |
|---|---|---|---|---|---|---|---|---|---|---|---|---|
| GAL | 1 | 0.75 | -0.66 | 0.95 | 2 | -1.08 | 0.65 | 3 | -0.66 | -0.86 | 0.85 | 1.5 |
| GUL | 1 | 0.75 | -1.08 | 0.95 | 2 | -1.37 | 0.75 | 2 | -1.08 | -1.37 | 0.95 | 1.7 |
| GLO | 1 | 0.8 | -0.66 | 0.75 | 1.5 | -1.08 | 0.57 | 2 | -0.66 | -0.86 | 0.65 | 1.5 |
| GAR | 1 | 0.85 | -1.08 | 1 | 3 | -1.37 | 0.8 | 4 | -1.08 | -1.37 | 0.95 | 2.2 |
| ROW | 1 | 0.9 | -0.66 | 0.87 | 2 | -1.08 | 0.62 | 3 | -0.66 | -0.86 | 0.87 | 1.2 |
| NYM | 1 | 0.85 | -0.86 | 1 | 1.25 | -1.08 | 0.65 | 3 | -0.86 | -1.08 | 1 | 1 |
| BEL | 1 | 0.8 | -1.08 | 0.95 | 1.5 | -1.37 | 0.65 | 3 | -1.08 | -1.37 | 0.95 | 1 |
| GAEL* | 1 | 0.75 | -1.08 | 1 | 1.5 | -1.37 | 0.9 | 1.5 | -1.08 | -1.37 | 1 | 1.3 |
| GLD | | | | 0.95 | 1.5 | | 0.95 | 1 | | | 0.65 | 3 |
| HES | | | | 1 | 1.2 | | 0.9 | 1.1 | | | 0.95 | 1 |

*fit with Mg/Camin substituted by Mg/Ca 5th percentile value of 6 mmol

## 4.5 Estimation of $\delta^{13}$C init in measured speleothems

Following the use of Scenarios A1, A2, and A3 to generate the fCa used in Equation 3, we generated time series records of
$\delta^{13}$C init for 8 stalagmites spanning three time periods of interest - the Late glacial to Holocene, the penultimate glacial to interglacial transition, and a stadial event occurring between 87 and 85 ka BP. For clarity in time series figures, we illustrate


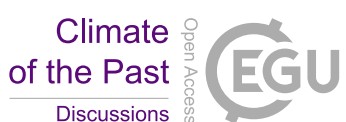

the three scenarios only for the -8‰ degassing slope. For one scenario (A1) we illustrate the range in $\delta^{13}C_{init}$ for degassing slopes from -5 to -11‰. This range is variable over time in a given stalagmite, because the choice of degassing slope (-5, -8, -11‰) impacts the $\delta^{13}C_{init}$ significantly for low fCa and to a lesser degree for higher fCa.

We begin by comparing the range of $\delta^{13}C_{init}$ calculated for growth during interglacials, and the present the time series trends for select individual stalagmites (time series for the remaining stalagmites are illustrated in the supplement).

### 4.5.1 Interglacials

The median $\delta^{13}C_{init}$ among interglacial samples (here averaging 9-4 ka; and 129 to 125 ka) is strongly dependent on the degassing fractionation slope A (Figure 6). For the three fCa scenarios (A1, A2, A3) in which Mg/Ca-based fCa is attenuated

for consistency with $\delta^{44}Ca$, with a -5‰ slope, median $\delta^{13}C_{init}$ ranges from -9 to -13‰ and for the -11‰ slope, $\delta^{13}C_{init}$ is in the -11.5 to -16‰ range. For scenario full, the -5 ‰ and -8 ‰ $\delta^{13}C_{init}$ overlap with modern nondegassed calcite predicted from $\delta^{13}C_{DIC}$ measurements, whereas for -11 ‰, scenarios A1 and full lead to $\delta^{13}C_{init}$ which is 5 to 7 ‰ more negative than modern $\delta^{13}C$ calcite of undegassed drip. Use of the "full" fCa scenario with no attenuation of the fCa from Mg/Ca and the slope of -11 ‰ leads to extreme negative $\delta^{13}C_{init}$ for GAR, GAL, GUL, and GAE but not BEL or GLD. With this fCa scenario, slopes

-5 or -8 ‰ yield $\delta^{13}C_{init}$ within the ranges described above.

### 4.5.2 Three coeval time series during MIS 5 b-c

Three stalagmites spanning 94 to 82 ka interval feature contrasting trends in measured $\delta^{13}C$ (Figure 7). An excursion to more positive measured $\delta^{13}C$ is noted around 89 to 86 ka BP in GLO, the stalagmite with the most constant Mg/Ca. However, the brief positive anomaly is not well in measured $\delta^{13}C$ GAE because of a high amplitude long term trend over this time interval.

Yet, in $\delta^{13}C_{init}$, all three stalagmites show a similar positive anomaly. The $\delta^{13}C_{init}$ with slopes -8 or -11, for all fCa scenarios, leads to a double structure positive $\delta^{13}C$ anomaly at 89 to 86 ka, and a constant background $\delta^{13}C_{init}$ from 94 to 82 ka. The $\delta^{13}C_{init}$ for these simulations in GAE exhibits a more similar trend to that of GLO and ROW.

A comparison of the range of fCa in each scenario, the resulting range in $\delta^{13}C_{init}$, and the correlation of measured $\delta^{13}C$ with the $\delta^{13}C_{init}$ is illustrated in Figure 7 panels f, g, and h. The most pronounced change between measured $\delta^{13}C$ and the $\delta^{13}C_{init}$

is simulated for GAE, in which correlation of $\delta^{13}C_{init}$ and measured $\delta^{13}C$ is below 0.2 for all degassing slopes except -5‰. The case of GAE contrasts with that of GLO, in which the correlation remains above 0.5 for all scenarios consistent with $\delta^{44}Ca$ (eg scenarios A1, A2, and A3). This reflects the greater range in fCa for GAE than for GLO.






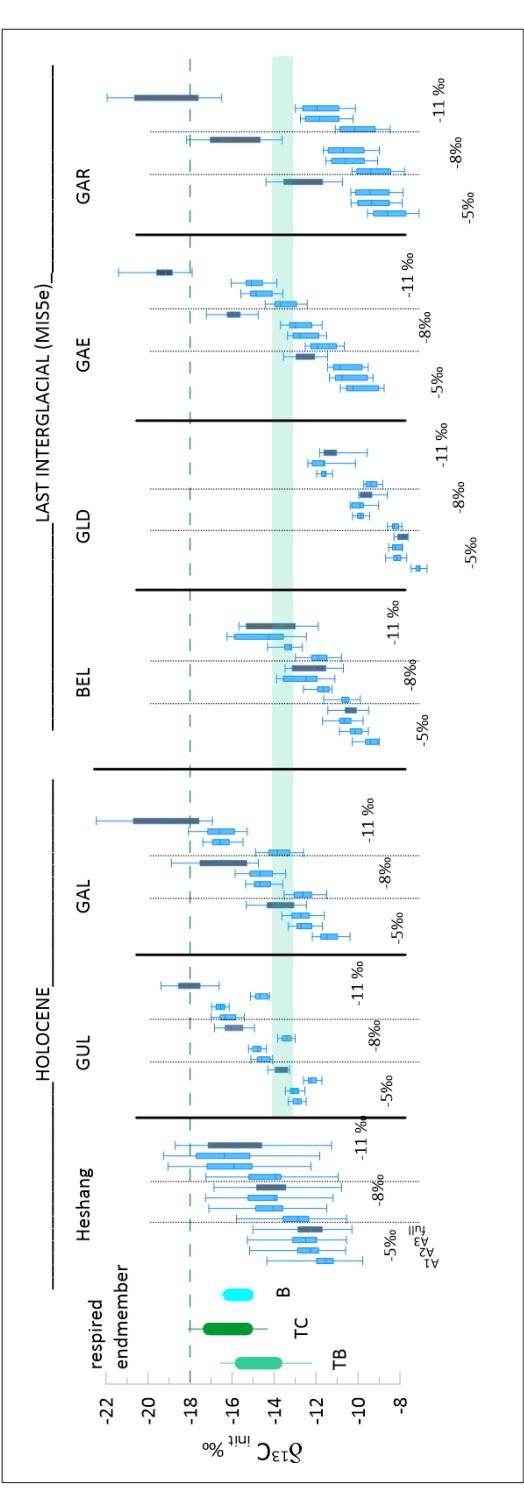

**Figure 6. Comparison of calculated $\delta^{13}C_{init}$ for interglacial samples spanning 9 to 5 ka and 129 to 125 ka with the calculated composition for calcite for modern $\delta^{13}C_{DIC}$ composition under forested portions of the cave (green horizontal band), as well as the calcite calculated for limited degassing of dripwaters equilibrated with soil pCO$_2$ (10000ppm) of composition consistent with the $\delta^{13}C$ of respired endmembers in temperate broadleaf (TB), temperate conifer (TC) and boreal (B) ecosystems (Pataki et al., 2003) illustrated as vertical ranges . For each stalagmite, the whisker plots show distribution of $\delta^{13}C_{init}$ for three fCa scenarios consistent between Mg/Ca and $\delta^{44}Ca$ (A1, A2, A3; blue shading) as well as the fCa derived from the full, unadjusted Mg/Ca record (gray shading) for three different degassing slopes. Shown is the median, upper and lower quartile, and with 99/1% whiskers for the four different fCa scenarios and three different slopes of degassing fractionation. The dashed green line gives the upper limit of calcite expected to form from dripwaters in the temperate conifer ecosystem; interglacial $\delta^{13}C_{init}$ not expected to be more negative than this value.**





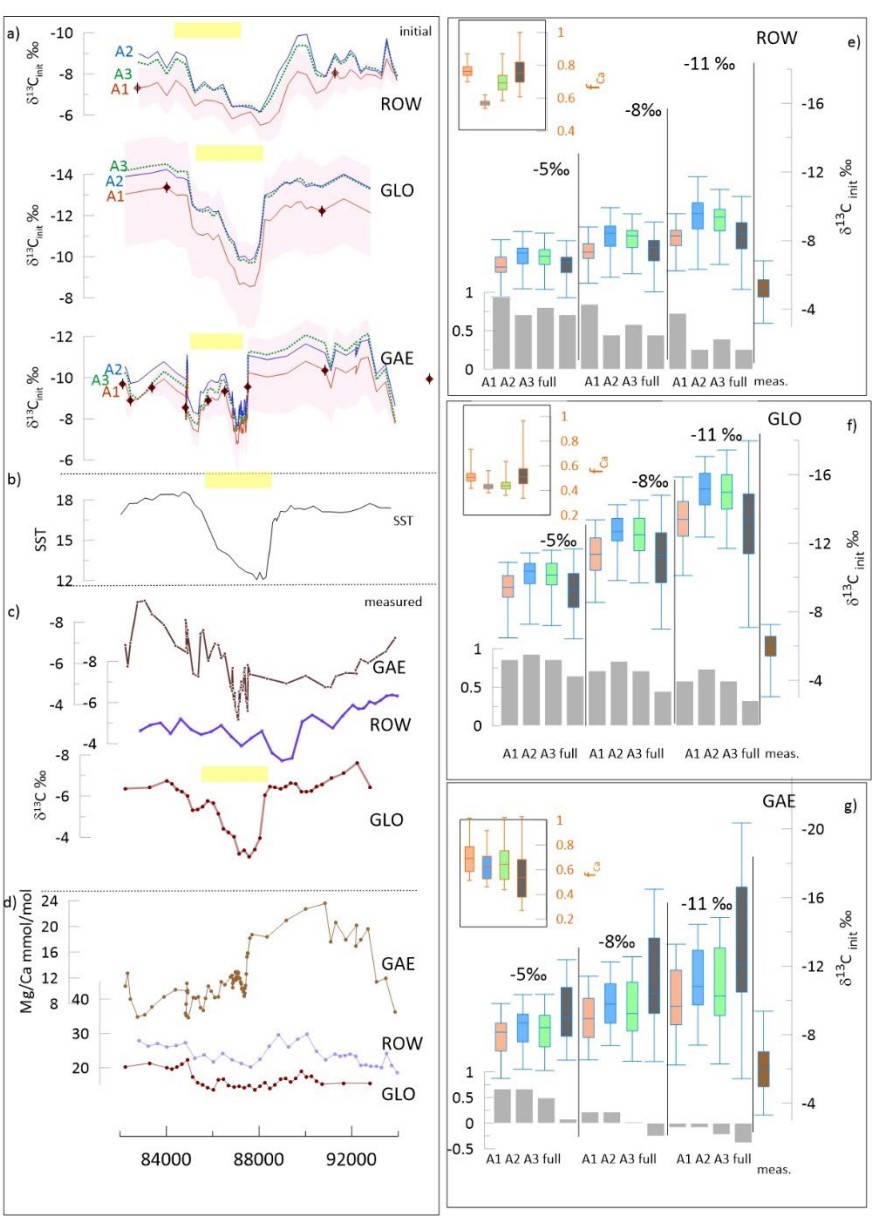

**Figure 7. Stadial event in time interval 94 to 82 ka in ROW, GLO, and GAE. a) for each stalagmite, the $\delta^{13}C_{init}$ for three fCa scenarios (A1, A2 solid lines, A3 dashed line); the line shows the result for degassing fractionation slope of -8 ‰, and shading for the A1 scenario shows the range using other degassing slopes of -5 ‰ (more positive), and -11 ‰ (more negative). Diamond symbols on the red curve indicate the position of U/Th dates. b) the SST record from the Iberian Margin (Martrat et al., 2007). c) the measured $\delta^{13}C$ for the three stalagmites, small symbols indicate measured samples. d) d) Mg/Ca for the three stalagmites, small symbols indicate measured samples. e) through g) inset shows the median and range of fCa for the various scenarios. Box and whisker plot shows the median, upper and lower quartile, and 1/99% ranges for the calculated $\delta^{13}C_{init}$ for each of the fCa scenarios (color coded as in Table 3 and Figure 5) and the measured $\delta^{13}C$. Gray bars at the base of each figure illustrate the Pearson correlation coefficient between the $\delta^{13}C_{init}$ and the measured $\delta^{13}C_{init}$. In a) through c), yellow bars highlight the inferred position of the stadial cooling event in each record, given uncertainty in age model.**





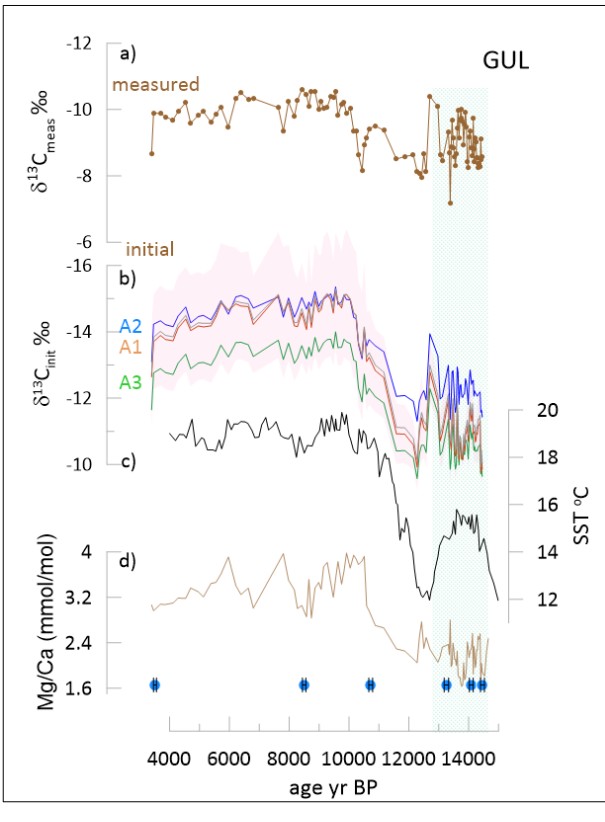

**Figure 8.** Glacial to Holocene transition in GUL with U/Th age control points illustrated along the age axis. a) the measured $\delta^{13}C$ for GUL. b) the $\delta^{13}C_{init}$ for three fCa scenarios; the line shows the result for degassing fractionation slope of -8 ‰, and shading for the A3 scenario shows the range using other degassing slopes of -5 ‰ (more positive), and -11 ‰ (more negative). c) the alkenone-based SST for the southern Iberian Margin(Cacho et al., 1999), d) the Mg/Ca for GUL. e) shows the median and range of fCa for the various scenarios. Box and whisker plot is given in Supplementary Figure 2. Green shading highlights the BA period.

### 4.5.3 Deglaciation

For the Holocene stalagmite GUL, there is a significant temporal trend of Mg/Ca increase from the Bølling Allerød (B/A) into the Holocene (Figure 8). Measured $\delta^{13}C$ was comparable in the B/A and the Holocene, but for all scenarios and degassing slopes, the $\delta^{13}C_{init}$ exhibits greater contrast between the early glacial and the Holocene, as the calculated $\delta^{13}C_{init}$ of the Bølling Allerød (B/A) is more positive than that of the Holocene.

A similar steepening of the temporal trend in $\delta^{13}C_{init}$ occurs for stalagmite GLD spanning the penultimate deglaciation. GLD measured $\delta^{13}C$ features a very stable average value punctuated

with two positive anomalies, while Mg/Ca exhibits a decrease over the deglaciation. The $\delta^{13}C_{init}$ features a significant negative shift between 136 and 130 ka. With this significant change in trends among $\delta^{13}C_{init}$ and measured $\delta^{13}C$, correlation coefficients drop well below 0.5 for two scenarios at the -8 slope of degassing and correlations approach 0 at the -11‰ slope (Figure 9).

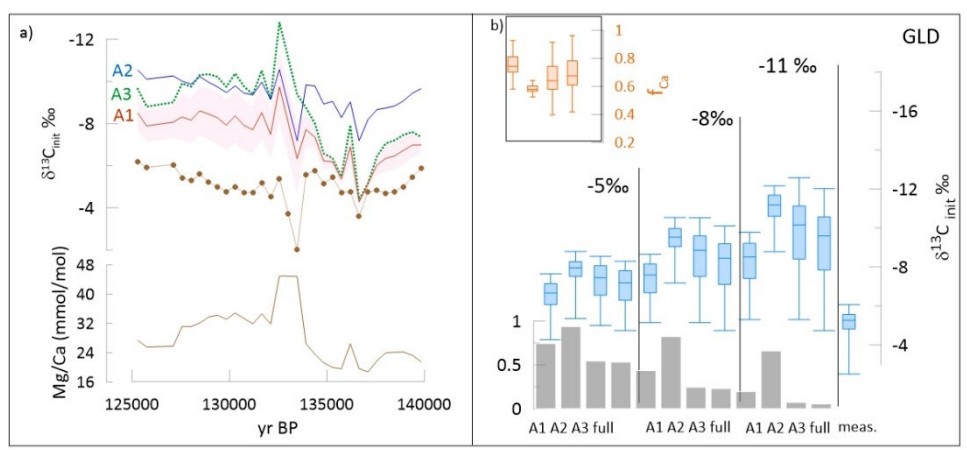

**Figure 9.** Penultimate glacial to interglacial transition in GLD showing the measured $\delta^{13}C$ (brown curve with symbols indicating measured samples) and colored lines the $\delta^{13}C_{init}$ for three fCa scenarios A1, A2, and A3 (dashed line) for degassing fractionation slope of -8 ‰. Shading for the A3 scenario shows the range using other degassing slopes of -5 ‰ (more positive), and -11 ‰ (more negative). Mg/Ca also shown with solid brown line. Box and whisker plot is given in Supplementary Figure 4.





## 5. Discussion:

**5.1 Interpretation of $\delta^{13}C_{init}$ and coherence of coeval speleothem signals and regional climate**

The calculated $\delta^{13}C_{init}$ is formulated to reflect the isotopic signal of DIC prior to significant evolution via degassing and PCP. Warmer interglacial climates characterized by higher soil $pCO_2$ promote both more negative $\delta^{13}C_{DIC}$ and increase the degree of dripwater saturation which promotes greater PCP for a given drip interval (Fig. 4). For some locations with drip intervals very sensitive to PCP, these processes may exert comparable magnitude but opposing influence on measured stalagmite $\delta^{13}C$

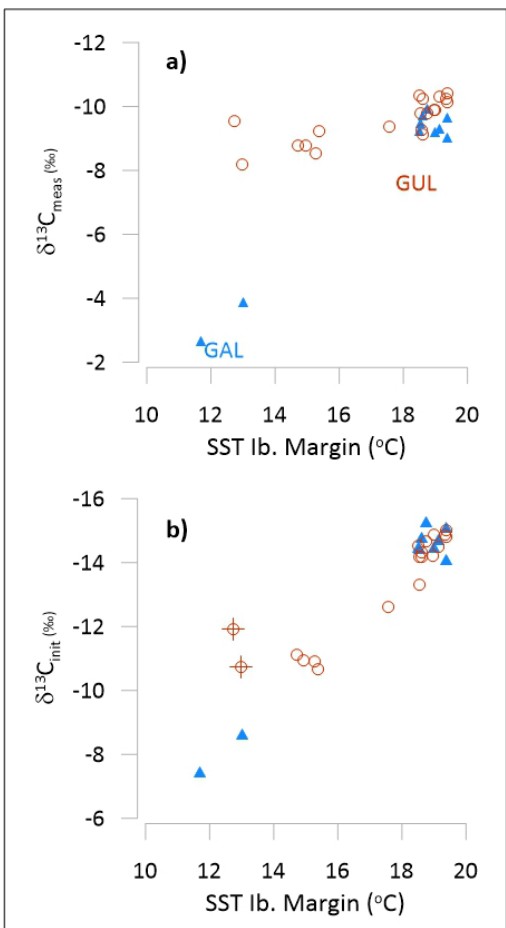

and lead to nearly constant measured $\delta^{13}C$ despite significant changes in temperature(Martrat et al., 2007) and vegetation(Goni et al., 2008; Tzedakis et al., 2018). We suggest that such process may have contributed to the similar average measured $\delta^{13}C$ in GUL in late glacial and Holocene time intervals (Fig. 6), and also to the similar glacial and interglacial measured $\delta^{13}C$ in GLD (Fig 7). Both of these stalagmites are characterized by large increases in Mg/Ca in the interglacial, indicating significant enhancements of PCP.

For stalagmite GUL growing during the last deglaciation, all of the examined calculations of $\delta^{13}C_{init}$ yield a temporal pattern more consistent with independent evidence for cooler regional temperatures during the late Glacial B/A compared to the early Holocene (Fig. 7) (Martrat et al., 2007; Ausín et al., 2019; Cacho et al., 1999). The aggregate $\delta^{13}C_{init}$ of both stalagmites spanning TI (GAL and GUL) exhibits a stronger correlation with regional SST than the measured $\delta^{13}C$ (Figure 10).

**Figure 10. Comparison in 500 ky fixed time bins of a) measured $\delta^{13}C$ and b) $\delta^{13}C_{init}$ vs. regional Iberian Margin SST (Cacho et al., 1999) over the last glacial to interglacial transition in GAL (blue triangle) and GUL(red circle). In b) we show a -8 ‰ degassing slope and scenario A1. Other scenarios for GAL illustrated in Supplementary Figure 3. In lower panel, crosses denote two samples of age 12.25 and 12.75 ka, where offset may reflect uncertainty in the age interpolation as shown in Figure 8.**

Because the extent of degassing and PCP can be strongly conditioned by the drip rate supplying an individual stalagmite, the extent and temporal variations of PCP may differ significantly among coeval stalagmites, leading to contrasting measured δ13C among coeval stalagmites even in the case of similar temperature and vegetation forcing of the initial dripwater $\delta^{13}C_{DIC}$. Between 94 and 82 ka, the Greenland Stadial (GS) 22 event is a salient cooling feature in regional SST records(Martrat et al., 2007). Yet, only one of the three stalagmites spanning this age range exhibits a clear positive anomaly in measured $\delta^{13}C$, stalagmite GLO which features stable Mg/Ca over this time interval. Stalagmite GAE features a long term trend towards





negative measured $\delta^{13}C$ not observed in GLO or ROW, and GAE features also a significant decrease in Mg/Ca (Fig. 7). The calculated $\delta^{13}C_{init}$ of GLO does not alter the trend seen in measured $\delta^{13}C$. However, the $\delta^{13}C_{init}$ of ROW features the positive

anomaly typical for the stadial cooling. Significantly, despite the strong long term trend in measured $\delta^{13}C$, the calculated $\delta^{13}C_{init}$ of GAE also resolves a positive anomaly consistent with the timing of the GS22 cooling. These more coherent temporal trends are resolved in the three records regardless of the fCa scenario employed.

These results for multiple coeval stalagmites suggest that calculation of $\delta^{13}C_{init}$ has the potential to improve reproducibility among coeval records and resolve signals of important regional changes in temperature and vegetation effects in the soil and

epikarst environment.

## 5.2 Coherency of quantitative PCP estimations and impact on deconvolving PCP effects in $\delta^{13}C$

### 5.2.1 Quantitative PCP indicators in this sample set

The calculation of $\delta^{13}C_{init}$ relies on quantitative estimation of PCP. PCP should exert a similar effect on ratios of divalent cations with low partitioning coefficients like Mg/Ca and Sr/Ca. However, many published Mg/Ca and Sr/Ca records do not

exhibit strong positive correlation(Sinclair, 2011). In our dataset, the greater correlation of Mg/Ca and Sr/Ca in some of the high Mg/Ca stalagmites may reflect the proportionally greater effect of calcite Mg/Ca on the $D_{Sr}$ in higher Mg/Ca stalagmites (eg Wassenburg et al 2019). A threefold reduction in Ca due to PCP is attainable for reasonable degrees of soil $pCO_2$ and dripwater saturation in our interglacial climates, and therefore a threefold range in Mg/Ca or Sr/Ca due to PCP is possible. Over a threefold range in Mg/Ca in low Mg/Ca stalagmites (eg 2 to 6 mmol/mol Mg/Ca), existing compilations suggest that

the $D_{Sr}$ increases by 20%. Yet, for a 3-fold increase in Mg/Ca in the high Mg/Ca stalagmites (e.g. 10 to 30 mmol/mol Mg/Ca) the $D_{Sr}$ would increase by 80%. Because Mg/Ca and $\delta^{44}Ca$ are always positively correlated, but Sr/Ca and $\delta^{44}Ca$ are not positively correlated in all samples, we conclude that growth rate-driven variations in $D_{Sr}$ can strongly overprint the PCP-driven trends in Sr/Ca in some stalagmites. For a given fCa, faster growth rates would promote a lower $\delta^{44}Ca$ and a higher DSr, fueling inverse correlation. Because Ca and Sr are dominantly bedrock sourced in this setting, this reversal from PCP-

expected relationship cannot be attributed to variable contribution of other element sources, but must reflect operation of processes at the solution-calcite interface. Similarly, deviation of Sr/Ca and Mg/Ca from the PCP slope, or variation in the Sr/Mg ratio does not conclusively require variation in non-bedrock sources in either element but could reflect partitioning effects which for Sr are decoupled from PCP.

In our dataset, temperature variation in DMg could explain some, but not all of the apparent amplification in the Mg/Ca-based

fCa compared to $\delta^{44}Ca$ -based fCa. Significant amplification is observed within the Holocene when temperature variation is limited. Additionally, in many cases the required range in DMg is larger than the 22% increase predicted from reasonable (eg <10 degree) amplitude temperature increases according to laboratory calibrations(Day and Henderson, 2013). One candidate explanation for higher magnitude changes in DMg may be the up to 2-fold increase in the DMg with increasing calcite Mg/Ca (Wassenburg et al., 2020). This effective partitioning increase may reflect true lattice partitioning and/or increased Mg in fluid



inclusions. The existing field data do not precisely constrain the slope of this increase in DMg or define if the slope is indeed linear. For future absolute estimation of fCa from Mg/Ca, improved constraints on the variation of DMg are required from laboratory and field studies. In field monitoring studies, true variation in DMg can be most accurately determined when observations adequately control for both the initial solution Mg/Ca and the integrated water Mg/Ca during calcite precipitation, which may be higher than the initial Mg/Ca if a significant extent of the Ca is precipitated on the stalagmite from each solution aliquot.

In addition to variation in DMg, temporal changes in the Mg/Ca of undegassed dripwater could also contribute to discrepancies between fCa from $\delta^{44}$Ca and Mg/Ca (e.g. the requirement for AF >1). We observe only a greater, never lesser, magnitude Mg/Ca change than that expected from changes in fCa alone. In our cave system, dripwater monitoring in La Vallina system shows very constant dripwater Mg/Sr ratios seasonally despite order of magnitude changes in dripwater flow rates (Kost et al., in review). This suggests that significant variation in the congruency of bedrock dissolution is not pervasive despite variation in water residence times.

### 5.2.2 Suggested approach for fossil stalagmites

Mg/Ca is widely measured PCP-sensitive indicator but can be unreliable in host rocks which feature significant components of widely varying solubility and Mg/Ca which give rise to strong incongruency of dissolution. Large variations in dripwater Mg/Sr ratios during monitoring may be one sign of such incongruency. Mg/Ca can also be unreliable as a PCP indicator if a large fraction of the Mg in the stalagmite is present in detrital minerals rather than in the calcite. Stalagmite Mg/Ca which covaries with Al/Ca or Si/Ca may indicate important detrital component of Mg. In these two situations, it is not recommended to use Mg/Ca for quantitative PCP estimation.

In settings where congruency is expected and detrital minerals are not a significant source of Mg in the stalagmite, the quantitative estimation of fCa nonetheless entails uncertainty. For stalagmites in which only Mg/Ca is measured, and $\delta^{44}$Ca is not determined, the constancy or range of variation in DMg will represent one source of uncertainty in the calculation of $\delta^{13}C_{init}$. The calculation of fCa from Mg/Ca can be adjusted by an attenuation factor as in Equation 7. An attenuation factor >1 is likely required if measured Mg/Ca leads to calculation of a larger fCa range than expected. For example, an fCa <0.20 would require initial dripwater Ca concentrations > 100 ppm. If a bedrock adjustment factor in Equation 7 is used, it may need to be coupled with an attenuation factor (other than 1) so that absolute fCa does not reach values lower than expected for the cave and climatic context. This uncertainty will be reduced as future research better constrains the variation of DMg with Mg/Ca.



### 5.3 What is the magnitude of PCP+degassing effect on $\delta^{13}C$?

### 5.3.1 Do interglacial stalagmites constrain the degassing $\delta^{13}C$ slope?

For interglacial stalagmites, we can estimate the lowest expected $\delta^{13}C_{init}$ and use this limit to evaluate the range of reasonable degassing slopes. The $\delta^{13}C_{DIC}$ of dripwater has been measured under forested sections of the cave, with sample collection techniques to limit degassing (Kost et al., in review). From this, we estimate the $\delta^{13}C_{init}$ for calcite forming under similar dripwater $\delta^{13}C_{DIC.}$ Additionally, we hypothesize that pre-anthropogenic land use at this location would have led to soil $pCO_2$ in the 8,000 to 10,000 ppm range during the mid-Holocene and previous interglacials(Lechleitner et al., 2021a). We therefore

calculate the composition of calcite forming from DIC in equilibrium with open system dissolution (eg 150 L gas volume and resulting 3% dcp) of limestone by 10000 ppm soil $CO_2$ of isotopic composition corresponding to modern respired endmembers in three ecosystems(Pataki et al., 2003)(Figure 6).

We find that interglacial stalagmites cannot rule out any of the investigated degassing slopes. The stalagmites from our monitored cave which include long Holocene growth phases are GAL and GUL, which require low to moderate attenuation

factors (1.3 to 3) for calculation of fCa from Mg/Ca. When the fCa calculated from Mg/Ca include the estimated AF for coherency with $\delta^{44}Ca$ as in Table 3, then the -8 ‰ slope for all GUL and GAL samples fall within the range of calcite composition expected from modern dripwater with limited degassing (Fig. 6). With the -11‰ slope, the median $\delta^{13}C_{init}$ approaches -17‰, which albeit higher than modern dripwater predictions, is still compatible with DIC deriving from equilibration with respired $CO_2$ from temperate broadleaf and conifer ecosystems(Pataki et al., 2003). With the -5‰ slope,

the median $\delta^{13}C_{init}$ is slightly below that implied for calcite on the basis of modern monitoring. Results for stalagmites spanning the last interglacial also indicate that the -8 to -11‰ slopes yield $\delta^{13}C_{init}$ coherent with expected interglacial calcite. BEL, which requires little attenuation, best matches expected range with the -11‰. But, our comparison does reveal that unreasonable corrections which are more negative than expected for the respired $CO_2$ component result from the combination of -11‰ degassing slope and fCa calculated from measured Mg/Ca without use of AF (scenario "full"), for some low Mg/Ca

stalagmites.

A similar analysis for the Holocene Heshang Cave, for which both Mg/Ca and fCa data are available, yields $\delta^{13}C_{init}$ with the -5‰ slope of -12‰ and with the -11‰ slope $\delta^{13}C_{init}$ of -15‰, both within the range of calcite composition expected from high $CO_2$ soils with respired $CO_2$ composition from temperature broadleaf or tropical ecosystems(Pataki et al., 2003) (Figure 6), consistent with the C3 evergreen broadleaf vegetation overlying Heshang Cave(Li et al., 2014).

Thus, the available interglacial datasets so far do not preclude the kinetically-enhanced slope of -11‰ for fractionation during degassing and PCP. In fact, for the last interglacial, for many stalagmites, the higher slopes yield $\delta^{13}C_{init}$ more consistent expected interglacial soil $pCO_2$. Although the degree of open vs closed dissolution cannot be inferred for last interglacial samples, if it is in the range of Holocene and late glacial samples from this and nearby caves (Lechleitner et al., 2021b), the





$\delta^{13}C_{init}$ is not expected to differ by more than (1‰) from that calculated for the relatively open 10000 ppm soil $CO_2$ CaveCalc
model.

### 5.3.2 Previous laboratory and field calibrations

The CaveCalc simulation assumes that the loss of carbon from dripwater is attained via $CO_2$ degassing and calcite precipitation
in a 1:1 ratio, with close coupling of the processes, and that fractionation is occurring at equilibrium according to laboratory-
estimated fractionation factors.

Empirical laboratory and field results are sensitive to true degassing and PCP as well as the isotopic consequences of exchange
between dripwater and cave atmosphere.

We are aware of one published field study in which dripwater was sampled before and after a ~3 m transit across a subvertical
flowstone surface in a cave with average temperature of 22.5°C (Mickler et al., 2019). In this transect, Ca and DIC were
directly measured and $\delta^{13}C$ DIC was determined from total evolution as $CO_2$ in flushed vials. Ca concentrations were on

average 20% lower after the ISST flowstone transit, and the median estimated degassing slope was -9.9‰ in the 8 monthly
paired samples. The slopes estimated on individual sampling dates (ranging from -7.7 to -13.7) did not correlate with dripwater
oversaturation nor the estimated ratio of Ca to DIC decrease on those dates.    The slope in this study is more than double that
of CaveCalc at comparable temperatures. One interpretation is that the rapid initial phases of degassing occur at kinetically-
enhanced fractionation factors (Mickler et al., 2019). A similar experiment conducted on one day in La Vallina Cave was also

consistent with a slope of -11‰, based on sampling during cave temperature of  at 12°C and modest saturation (55 ppm Ca in
dripwater, vs 21 ppm for equilibrium at cave temperature and $pCO_2$  ppm), (Kost et al., In review). During this sampling,
cave air was ventilated to near atmospheric $CO_2$ concentrations and isotopic composition. These laboratory and field results
suggest that, given current available data, the CaveCalc slopes may need to be considered the minimum slopes required to
correct for degassing and PCP effects on carbon isotopes.

The laboratory experiment best simulating the coupled evolution of $\delta^{13}C$ DIC, calcite   $\delta^{13}C$ and fCa allowed highly
supersaturated solutions to degas and precipitate $CaCO_3$ as they flowed along a marble plate, with electrical conductivity
measurements estimating the degree of Ca depletion along the flow path in a parallel experimental setup with flow over a glass
plate (Hansen et al., 2019). The $\delta^{13}C$ of DIC was measured by precipitation in $SrCO_3$ and analysis as solid phase. The slope
A in the 10 experiments, based on the evolution of $\delta^{13}C$ of DIC, ranged from  -4.63 to -13.2‰, or from -5.9 to -16‰ based on

evolution of $\delta^{13}C$ of calcite, although precise estimation of the latter slope is complicated by the measurement of precipitated
calcite $\delta^{13}C$  and conductivity in separate experimental setups which may have experienced slightly different locations of
calcite precipitation(Sade et al., 2022). Many of these estimated experimental fractionations are considerably higher than those
predicted by CaveCalc.

A dynamical model (Sade et al., 2022)  was developed to simulate the laboratory experiments(Hansen et al., 2019).  This

model derived a single relationship between PCP  and DIC evolution, using fDIC in contrast to our calculated index based on




Ca comsumption, fCa. In this model formulation, the degassing slope relative to ln (fDIC) averages -8.7 ‰ from f=1.0 to f=0.3, but the slope is not constant. Rather, at fDIC between 0.8 and 1, there is an inverse slope as net DIC evolution is modelled to lead to more negative, rather than more positive, $\delta^{13}C$ of speleothem calcite. Over the subsequent phase of PCP (fDIC 0.8 to 0.3), the average modelled degassing slope relative to fDIC would be -10.1‰, however it flattens to -4‰ in the range of fDIC 0.33 to 0.4. If this model is representative of cave flowstone formation, it may suggest that some of the calculated differences in degassing slopes in field experiments may result from measurements of solutions sampled in different stages of dripwater Ca and DIC evolution. Alternatively, it is possible that some differences in degassing slopes reflect different degrees of coupling between degassing and $CaCO_3$ precipitation. The influence of factors inhibiting $CaCO_3$ precipitation, potentially including elevated Mg concentrations or certain dissolved organic compounds, on the degassing slope remains to be investigated.

If there were significant variations in the degassing slope experienced by a given stalagmite over time, it could complicate efforts to estimate trends in $\delta^{13}C_{init}$ using Eqn. 2. Yet, our speleothem time series are broadly consistent with narrow range of variation of degassing slope within a given stalagmite, since correction of -5 ‰ in one part of the time series and -11‰ in another would alter the trends and reduce coherence among the records in Figure 7. This coherence with a single slope may reflect the limited range of variation in fCa inferred for our stalagmites. For example, for all stalagmites except GAE, the A2 scenarios, and the majority of data from A3 scenarios, fall between fCa of 0.4 and 0.8. If coincident with fDIC of 0.4 to 0.8, these samples would all be within the range of similar slope (-10 to -11.5‰) predicted by the dynamical model simulation. Sampled stalagmite growth may often be biased to periods when fCa >0.4, which lead to higher precipitation rates and more calcite deposition. The consistency of our interglacial data with steeper degassing slopes than CaveCalc may reflect the reality that much calcite precipitation will happen when cave $pCO_2$ is much lower than dripwater $pCO_2$, a situation in which dripwater exchange with isotopically heavier cave air could be significant contributor to a steep degassing slope.

A future calculation of the dynamical model prediction(Sade et al., 2022) of $\delta^{13}C$ evolution relative to fCa, rather than fDIC, as a PCP indicator, would provide a helpful reference to future calculation of $\delta^{13}C_{init.}$ This could elucidate whether a nonlinear relationship should be used to calculate $\delta^{13}C_{init}$ from ln (fCa) and $\delta^{13}C_{meas}$. It would clarify if calcite precipitated at high fCa, analogous to that precipitated at high fDIC, should be excluded from the correction because of compensating effects of kinetic fractionation factors and evolution of fCa in the range of high fCa. Scenarios employing a B ≤0.8 by default have a maximum fCa largely outside the range of inverse slope.

### 5.3.3 Suggested approach

Trends in $\delta^{13}C_{init}$ are less sensitive to the choice of the degassing slope but the absolute value of $\delta^{13}C_{init}$ is strongly dependent on the choice of degassing slope. At the moment, the absolute $\delta^{13}C_{init}$ can be reconstructed with low confidence. However, the reconstructed $\delta^{13}C_{init}$ can be used to rule out combinations of fCa scenarios and degassing slopes which lead to



"overcorrection" to $\delta^{13}C_{init}$ values which are more negative than DIC equilibrated with a reasonable respired soil $pCO_2$ composition for the ecosystem type. Until further constraints to assess the slope of $\delta^{13}C$ and Ca co-evolution during degassing

and PCP which characterizes typical cave environments are available, a sensitivity analysis employing a range of degassing slopes will provide the greatest transparency for assessing $\delta^{13}C_{init.}$

## 6. Conclusions

We provide a first analysis of the potential estimation of speleothem $\delta^{13}C_{init}$, the $\delta^{13}C$ which characterized dripwater DIC prior to significant degassing and PCP. $\delta^{13}C_{init}$ is proposed as a useful interpretable variable derived from speleothem isotope

measurements, because trends in $\delta^{13}C_{init}$ are expected to be more reproducible than measured $\delta^{13}C$ among coeval stalagmites from a given region and because $\delta^{13}C_{init}$ is more sensitive to the vegetation, soil, and epikarst processes which in many regions may be sensitive to temperature in addition to moisture. Calculation of $\delta^{13}C_{init}$ for a given sample or growth increment requires a quantitative indicator of the extent of PCP experienced in that growth increment , and knowledge of the general rate of change of $\delta^{13}C$ with progressive degassing and PCP.

In fossil stalagmites, the extent of PCP, as the Rayleigh fCa, can be approximated for a particular growth increment using the measured Mg/Ca and the minimum Mg/Ca measured in the entire stalagmite. fCa can also be estimated from $\delta^{44}Ca$ given reasonable choices of the calcite-dissolved fractionation factor. In several of the stalagmites examined here, the fCa calculated from Mg/Ca shows an amplified range compared to that calculated from $\delta^{44}Ca$. An increase in DMg with increasing Mg/Ca, as proposed in previous studies (Wassenburg et al., 2020), is one viable explanation for this systematic trend. This observation

warrants further investigation to improve confidence in future estimates of fCa from Mg/Ca, which is the most widely available PCP indicator for stalagmites.

At the moment, there is uncertainty in the degassing slope or rate of change of $\delta^{13}C$ with progressive degassing and PCP. Values may range from -5 to -11‰. The stalagmites investigated here could be consistent with any value in this range. Because these different possible degassing slopes affect the absolute value of the calculate $\delta^{13}C_{init}$, currently there is uncertainty

in exact value. However, the trends in $\delta^{13}C_{init}$ are less sensitive to the choice of degassing slope.

Despite these uncertainties, $\delta^{13}C_{init}$ provides more consistent time series among coeval stalagmites and with regional climate records. In one example $\delta^{13}C_{init}$ reconciles three divergent measured stalagmite $\delta^{13}C$ records in the 94 to 82 ka time interval, yielding three $\delta^{13}C_{init}$ time series which feature a pronounced positive anomaly corresponding to the regional cooling of Greenland Stadial 22. In Western Europe, over the warming trend of deglaciations, the trend towards more negative $\delta^{13}C_{DIC}$

due to higher soil respiration and soil $CO_2$ may be fully or partially compensated in stalagmite $\delta^{13}C$ due to the increased PCP from the more oversaturated dripwaters. The calculation of $\delta^{13}C_{init}$ reveals the trend of increasing respiration rates and soil $pCO_2$. Over the last deglaciation, the $\delta^{13}C_{init}$ matches millennial structure in regional SST more closely than the measured $\delta^{13}C$.



As better constraints emerge on the degassing slope and on Mg partitioning, $\delta^{13}C_{init}$ estimates will become more precise and
should improve the utility of the high volume of stalagmite $\delta^{13}C$ measurements made simultaneous with $\delta^{18}O$ in all labs.

Appendix A. Measured $\delta^{44}Ca$, Mg/Ca and Sr/Ca, and the calculated fCa for differing $\Delta^{44}Ca$, and B (bedrock factors).

| | $\delta^{44/40}Ca$ ‰ | Mg/Ca (mmol/mol) | Sr/Ca (mmol/mol) | fCa from $\delta^{44}Ca$ $\Delta$44Ca ‰ | | | | fCa from Mg/Ca B | | | fCa from Sr/Ca B | | |
|---|---|---|---|---|---|---|---|---|---|---|---|---|---|
| | | | | -0.66 | -0.86 | -1.08 | -1.37 | 1 | 0.8 | 0.6 | 1 | 0.8 | 0.6 |
| GAL-37.5 | 0.21 | 3.12 | 0.034 | 0.64 | 0.56 | 0.52 | 0.48 | 0.35 | 0.28 | 0.21 | 0.69 | 0.55 | 0.41 |
| GAL-42 | 0.19 | 2.78 | 0.035 | 0.66 | 0.58 | 0.53 | 0.49 | 0.39 | 0.31 | 0.24 | 0.66 | 0.53 | 0.40 |
| GAL 42.5 | 0.12 | 1.79 | 0.035 | 0.74 | 0.63 | 0.562 | 0.51 | 0.61 | 0.49 | 0.36 | 0.67 | 0.53 | 0.40 |
| Gal 16 | 0.06 | 1.57 | 0.040 | 0.80 | 0.67 | 0.59 | 0.54 | 0.70 | 0.56 | 0.42 | 0.57 | 0.46 | 0.34 |
| Gal 30 | 0.14 | 2.67 | 0.032 | 0.72 | 0.61 | 0.553 | 0.51 | 0.41 | 0.33 | 0.25 | 0.73 | 0.58 | 0.44 |
| Row 452 | 0.04 | 19.06 | 0.105 | 0.84 | 0.69 | 0.61 | 0.55 | 0.95 | 0.76 | 0.57 | 0.90 | 0.72 | 0.54 |
| Row 252 | 0.12 | 26.58 | 0.129 | 0.73 | 0.63 | 0.56 | 0.51 | 0.68 | 0.54 | 0.41 | 0.74 | 0.59 | 0.44 |
| BEL mid 210 | -0.40 | 9.56 | 0.109 | 1.62 | 1.15 | 0.91 | 0.75 | 0.95 | 0.76 | 0.57 | 0.69 | 0.56 | 0.42 |
| BEL mid 60 | -0.13 | 14.52 | 0.130 | 1.08 | 0.84 | 0.71 | 0.62 | 0.62 | 0.50 | 0.37 | 0.58 | 0.46 | 0.35 |
| GAE_33.5 nd | -0.41 | 4.39 | 0.109 | 1.66 | 1.17 | 0.92 | 0.76 | 0.49 | 0.39 | 0.29 | 0.23 | 0.18 | 0.14 |
| GAE_60.4 nd | 0.13 | 17.59 | 0.115 | 0.73 | 0.62 | 0.56 | 0.51 | 0.12 | 0.10 | 0.07 | 0.22 | 0.17 | 0.13 |
| GAR_b2_035.0 | -0.25 | 7.42 | 0.097 | 1.29 | 0.96 | 0.79 | 0.67 | 0.33 | 0.26 | 0.20 | 0.29 | 0.23 | 0.17 |
| GAR_b6_131.5 | -0.49 | 2.42 | 0.073 | 1.85 | 1.27 | 0.99 | 0.80 | 1.00 | 0.80 | 0.60 | 0.39 | 0.31 | 0.23 |
| GLO_b1_17.4 | 0.23 | 9.74 | 0.037 | 0.63 | 0.55 | 0.51 | 0.48 | 0.78 | 0.62 | 0.47 | 0.55 | 0.44 | 0.33 |
| GLO_b2_08.2 | 0.42 | 17.24 | 0.068 | 0.47 | 0.44 | 0.43 | 0.41 | 0.44 | 0.35 | 0.26 | 0.30 | 0.24 | 0.18 |
| GUL4-2.3 | -0.15 | 2.19 | 0.062 | 1.11 | 0.86 | 0.72 | 0.63 | 0.74 | 0.59 | 0.45 | 0.41 | 0.33 | 0.25 |
| GUL_I-3 | 0.06 | 3.69 | 0.033 | 0.81 | 0.67 | 0.60 | 0.54 | 0.44 | 0.35 | 0.27 | 0.78 | 0.62 | 0.47 |
| NYM_331 | -0.13 | 11.29 | 0.119 | 1.08 | 0.84 | 0.71 | 0.62 | 0.82 | 0.66 | 0.49 | 0.74 | 0.59 | 0.45 |
| NYM_457.5 | -0.04 | 13.67 | 0.134 | 0.94 | 0.75 | 0.65 | 0.58 | 0.85 | 0.68 | 0.51 | 0.69 | 0.56 | 0.42 |

**Data availability.**

Upon acceptance, data will be archived on the ETH data repository on the SISAL template, and presented for inclusion in the
subsequent version of SISAL.

**Supplementary file** accompanies this manuscript.

**Author contributions.** HMS conceived the study, conducted calculations, prepared figures and wrote the text with discussions
from CD and FL. CD completed Ca isotope measurements. FL and LE conducted CaveCalc simulations. OK and JS assisted
with trace element analysis and sampling for Ca isotopes and dating. DS contributed to interpretation. HC and CP completed
new chronology.



**Competing interests.**

The authors declare they have no conflict of interest.

**Acknowledgments**

HS acknowledges ETH core funding and grant ETH-13 18-1. FL was supported by SNSF grant P400P2_180789. CD and Ca-isotope measurements were supported with funds from John Fell Oxford University Press Research Fund grant 0007911. FL
was supported by SNSF grant P400P2_180789. CP and HC acknowledge NSFC (41888101, 42050410317) and Postdoctoral Science Foundation of China (2020M683452). We thank ETH Climate Geology lab manager Madalina Jaggi and student assistants Tim Loeffel and Romain Alosius for assistance with stable isotope and trace element sampling. We thank the ETH fall 2021 Paleoclimate course students and course assistant Pien Anjewierden for preparing samples from GLD, and Fall 2019 Paleoclimate course students for initiating study of GUL.

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
