# Peer review of "Distinguishing the vegetation and soil component of $\delta^{13}$ C variation in speleothem records from degassing and prior calcite precipitation effects"

_Climate of the Past, 2022_

## Author Comment (AC1)

**Response to Reviewer Comment 1 (reviewer comment in black, response in violet and revised text in blue).**

The manuscript "Distinguishing vegetation and soil component of d13C variation in speleothem records from degassing prior calcite precipitation effects" by Stoll et al. presents a high quality scientific contribution to the interpretation of speleothem proxy records, in particular carbon isotopes, as well as the trace metal ratios (Mg/Ca) and (Sr/Ca) also explored in the manuscript. It is within scope of Climate of the Past for the development of methods and understanding of speleothem proxy data, as well as some new data in the case studies. The results and interpretation well-argued and presented in an appropriately structured way in terms of text and figures/tables.

The approach is novel and the methods valid. The manuscript is particularly well-presented in its presentation of the complexity of carbon isotopes and PCP, providing a useful review of these proxies as well as a novel method towards improved quantification of proxies. The paper also well-outlines the uncertainties in the approach and then develops a method to numerically incorporate these uncertainties into the quantification of the impact of degassing/PCP on speleothem carbon isotopes. It is shown that while the uncertainties preclude a reconstruct an actual soil CO2 d13C value, it is shown that the method may still be applied to examine the impact on the trend of the reconstructed soil δ13C compared with the measured δ13C values in the speleothem. This is the strength of the paper and is well-illustrated in the case studies. I appreciated the detailed appraisal of the method and output.

As well as providing a method for correcting the PCP impact on d13C via Mg/Ca, my impression of other strengths of this paper are also is in its demonstration that speleothem DMg may vary within a speleothem time series and that this may actually be expected to be common. I think that this point is an important contribution to the community and it is well-explored in the paper, although please see point below about fabrics and flow paths. The other major finding that I think should be useful to the community is that the soil PCO2 and PCP processes can have counteracting influences on the resulting stalagmite δ13C and produce observed nearly constant δ13C timeseries during significant climatic transitions. It is demonstrated in the two case studies and provides good motivation for others to investigate their datasets with this method.

There appears to be cave monitoring paper from the same cave(?) in review (Kost et al) that this was not cited in the reference list. If it were (and if in open access review?), it would have been interesting to see whether these monitoring data support the approach presented here. Hopefully, this is covered in the Kost paper.

The Kost et al. monitoring study, in the review process since 2021, is now published and the full citation will be included in the revised manuscript. It is:

Kost, Oliver, Saul González-Lemos, Laura Rodríguez-Rodríguez, Jakub Sliwinski, Laura Endres, Negar Haghipour, and Heather Stoll. "Relationship of seasonal variations in drip water δ 13 C DIC, δ 18 O and trace elements with surface and physical cave conditions of La Vallina Cave, NW Spain." *Hydrology and Earth System Sciences* v 27 2023 (2022): 1-42. https://doi.org/10.5194/hess-27-2227-2023.

As outlined above, the authors have well-outlined the uncertainties in their approach, although there are two other possible processes that were not or perhaps under-acknowledged. One is the the recent findings by Frisia et al 2022 in Quaternary Science Reviews. This paper showed that Mg/Ca in the calcite may vary with fabric porosity. This will affect the partition coefficient, DMg, which feeds into the equations for correcting PCP impact on d13C. This should be included and the findings of Frisia et al paper also implies

that the Stoll et al manuscript would benefit from a description of the fabrics and whether they differ within a stalagmite over sections where DMg is thought to have varied.

We appreciate the reviewer alerting us to this new paper. We propose adding the citation to Frisia et al 2022, which was published after our manuscript submission, in section 5.2.1.

Future field monitoring and farmed calcite studies should also evaluate whether calcite fabrics could provide independent evidence for variation in DMg, since some stalagmites show relationship between fabrics and Sr partitioning (Frisia et al., 2022).

In the background information on the samples, we propose to provide information on fabrics for several stalagmites where such data has been previously published:

The slowly growing fossil stalagmites consist of dense calcite and show no evidence for columnar porous fabrics. Stalagmite calcite for GAR and GAL consist of predominantly columnar compact to columnar open fabrics (Sliwinski and Stoll, 2021). Similar fabrics are confirmed in GUL and GAE(Sliwinski et al., 2023). Samples ROW, BEL, and NYM fall exhibit growth rates (12-40 $\mu$m/yr) similar to GAR, GAL, GUL, and GAE and feature similar dense, nonporous macroscopic textures. The much more slowly growing GLO and GLD likewise feature dense, compact calcite comparable to sample GLA with similar growth rate and described as columnar and columnar open (Kost et al., in review).

This includes a citation to the following paper, which presents multi-technique examination of fabrics and trace element maps on active and fossil samples from the same cave setting We will provide updated citation when its review process is complete.

Kost, O., Sliwinski, J., Gies, N., Lueder, M., and Stoll, H. M.: The influence of fluid inclusions, organics, and calcite fabric on trace element distributions in stalagmites, Frontiers in Earth Science, in review.

The potential for speleothem fabrics to provide further clues on changing partitioning coefficients is one which clearly merits further study, because we identify the potential for variation in DMg to be a key uncertainty in quantitative estimation of PCP. Our results comparing $\delta^{44}$Ca and Mg/Ca suggest that the DMg may be systematically increased as dripwater Mg/Ca increases, ie DMg may covary with PCP in a given stalagmite. In the revision in section 5.2.1, we propose to clarify that these relationships should be carefully investigated in future experimental and cave monitoring studies, in which the DMg can be independently calculated. In fossil stalagmites, in which the dripwater composition is not independently constrained, we expect that it could be difficult to ascertain the causes and consequences of correlations among PCP, fabrics, growth rate, and DMg. Also considering the suggestion of Reviewer 2, that the manuscript and its figures are currently too long and should be reduced, we refrain from introducing further complexity in this paper with addition of fabrics, and underscore that this is an important avenue of investigation for future work with the following statement:

Future field monitoring and farmed calcite studies should also evaluate whether calcite fabrics could provide independent evidence for variation in $D_{Mg,}$ since some stalagmites show relationship between fabrics and Sr partitioning (Frisia et al., 2022).

Similarly, karst flow paths received one brief mention around line 310 yet there has been much discussion in the literature around flow paths and PCP. The manuscript should include a few more sentences around this, rather than focus on PCP has just being a function of CO2 gradients and drip rate. Finally, the fabrics and even speleothem morphology can provide a useful indication of past drip rates and this is covered in multiple other studies and in the Speleothem Science book by Fairchild & Baker. Seeing as the influence of drip rate on PCP appears to be important yet we cannot know how drip rate has varied in the past, some description of the fabrics and speleothem morphology could have been helpful as these can be an indication on whether drip rate has varied in the past.

We thank the reviewer for these suggestions and propose to take up karst flow paths in more detailed discussion of PCP. We propose expanding the second paragraph of section 4.4.1 to clarify this issue:

In addition to the oversaturation state, PCP is also dependent on the degassing time. PCP can occur in air-filled voids above the cave as well as on walls and ceilings of the cave prior to the landing of dripwater on the studied stalagmite. Here, we discuss the integrated PCP, regardless of where along the flow path PCP has occurred. The susceptibility of a given speleothem to PCP may depend on the geometry of the flow path. Temporal variations in PCP in a given stalagmite are expected to depend on the flow (e.g. drip rate) as well as on the oversaturation.

Other points:

More correct wording of the title may be: "Distinguishing soil component of d13C variation in speleothem records from degassing prior calcite precipitation effects". The approach is not able to specifically isolate vegetation d13C from the soil d13C CO2 pool and does not discuss vegetation-derived d13C CO2 at any length in the manuscript.

We thank the reviewer for the suggestion to clarify the title. We proposed to distinguish the joint soil and vegetation processes from the degassing/PCP process, we do not propose that further deconvolution can be made to separate soil effects from the vegetation of the soil and epikarst. For this reason our target variable is described as the $d13C_{init}$, the isotopic value of DIC after soil and vegetation interaction, but prior to degassing and PCP.

We propose that a clearer title may be adding the word combined to clarify that the initial d13C signal is set jointly by the vegetation and soil processes:

**Distinguishing the combined vegetation and soil component of δ13C variation in speleothem records from subsequent degassing and prior calcite precipitation effects**

We also propose in the introduction, to clarify

This soil and vegetation signature imparted to the dripwater is also imprinted on speleothem $\delta^{13}C$.

L19: the term "overprinting" is used to describe PCP on speleothem $\delta^{13}C$ here and throughout. Suggest "contribution" would be more correct term.

We believe it is helpful to distinguish between the initial signal attained by dripwater equilibration with $CO_2$ from soil and deeper respiration, a signal we refer to as $\delta^{13}C_{init}$, from the subsequent evolution of this signal due to degassing. We selected overprinting as it conveys the generally sequential process, and suggest to retain this term.

L22: "universally", this is not a global study so the conclusion can only be that PCP is not the dominant control on this particular study site. Suggest reword.

We thank the reviewer for the suggestion to reword, which we will implement.

L28: this sentence is unclear and needs rewording, suggest: "During glaciations, calculated initial δ13C implies trend of increasing respiration rates and higher soil CO2, despite the interpreted reduced drip flux to favour more extensive PCP"?

We thank the reviewer for the suggestion to reword and clarify, which we will implement as follows:

During deglaciations, calculated initial $\delta^{13}C$ implies a trend of greater respiration rates and higher soil $CO_2$, although the higher interglacial dripwater saturation favors more extensive PCP.

L32: Why not the lower latitudes? Doesn't Figure 4a contradict this introductory sentence?

We thank the reviewer for the suggestion to reword and clarify, which we will implement. We agree that changes in vegetation productivity and soil processes significantly influence the $\delta^{13}C_{init}$ across all latitudes. Now, we have added the focus on mid and high latitudes to the fourth sentence instead, where we discuss the climate sensitivity of respiration rates, to emphasize that only in the mid to high latitudes is temperature likely to limit respiration rates. This paragraph then would read:

Changes in vegetation productivity and soil processes significantly influence the $\delta^{13}C$ of dripwater. The $\delta^{13}C$ of $CO_2$ in the soil and karst reflects the relative contributions of isotopically light respired $CO_2$ and isotopically heavier atmospheric $CO_2$. Conditions which favor higher vegetation productivity and faster rates of heterotropic and autotrophic respiration in soils will lead to higher soil $pCO_2$ and a lower $\delta^{13}C$ of $CO_2$. In contrast, in less productive and slower respiring systems, the atmospheric $CO_2$ and its higher $\delta^{13}C$ will be more significant C sources in the soil. This soil and vegetation signature imparted to the dripwater is also imprinted on speleothem $\delta^{13}C$. In the temperature range characterizing the mid- and high latitudes, respiration rates and vegetation density are highly sensitive to temperature, and speleothem $\delta^{13}C$ has been exploited to serve as a temperature proxy in mid-latitude speleothems(Genty et al., 2006; Genty et al., 2003).

L39: root depth and soil moisture content also contribute.

We thank the reviewer for the suggestion to reword and clarify. We have added the reference to Meyer et al 2014 here, which discusses the significance of deep-rooted trees to maintaining a high column-integrated CO2 and open system dissolution. We also note

the role of sufficient soil moisture in maintaining high soil $CO_2$ citing Romero-Muialli et al 2019 .

Table 1: looks like a copy and paste mistake for scenario A1 and A2 as identical definitions given for these variables. This made it a little hard unfortunately to understand some of the distinction in the modelled results for these two scenarios although I don't think was an important factor in being able to follow the main findings of the paper. But needs fixing here. Also, Delta Ca should be Delta44Ca.

We thank the reviewer for suggestions to clarify the table 1 notation. There is NOT a copy and paste mistake regarding scenario A1 and A2 in Table 1. Both scenarios A1 and A2 employ constant $\Delta$Ca, but with a different choice of its value. For each stalagmite, the precise value of $\Delta$Ca in A1 and A2 is specified in Table 3. We propose to adjust the table to provide a single line for scenario A1 and A2 so that they share the common description. We will also adjust the notation of $\Delta$44Ca

L167: suggest precipitation rate rather than growth rate dependent, i.e., the loss of calcium is not necessarily related to growth rate. Growth rate is usually used to define the linear extension rate in the vertical axis of a stalagmite.

We retain the term growth rate in the discussion of the fractionation factors, since growth rate is the term employed in the cited DePaolo, 2011 and Mills et al 2021 references.

Sub-heading 2.2.3. Are there a few more papers on D44Ca and PCP worth citing here?

In this section 2.2.3, we cite only studies which have presented both Mg/Ca and d44Ca on paired samples, in order to compare the results from both indicators. We have added a subsequent study on Heshang Cave (Li et al., 2018) to the cited Owen et al study (2016). Other studies reporting d44Ca but not Mg/Ca are cited elsewhere, including De Wet et al 2021 (cited in section 2.2.2 introducing Ca isotopes as PCP indicator).

L216-217: is this statement supported by the back modelling in the results?

We are now able to update the citation here for monitoring results, which provide full support for the statement

Kost, Oliver, Saul González-Lemos, Laura Rodríguez-Rodríguez, Jakub Sliwinski, Laura Endres, Negar Haghipour, and Heather Stoll. "Relationship of seasonal variations in drip water δ 13 C DIC, δ 18 O and trace elements with surface and physical cave conditions of La Vallina Cave, NW Spain." *Hydrology and Earth System Sciences* v 27 2023 (2022): 1-42. https://doi.org/10.5194/hess-27-2227-2023.

L222: please add the mass spectrometer specifications.

We infer the reviewer is suggesting that the detail the analytical reproducibility provided in the Breitenbach and Bernasconi reference. We will add the precision is 0.08 ‰ for both isotopes.

L231: please reword as it is unclear as to whether there were five repeats on each aliquot or whether there were five analyses performed on each stalagmite?

We now clarify that a minimum of 5 analyses were conducted on each aliquot.

L240: preferred value rather than preferred measurement?

We refer to Mg/Ca as the preferred measurement relative to $\delta^{44}Ca$. We have adjusted the sentence for greater clarity.

L253: please refer back to equation 3 here.

We will add the reference to equation 3.

L300: La Vallina Cave?

Thank you for prompting us to clarify that we refer to La Vallina Cave, in Porrua.

L300 paragraph starting here becomes confusing as total range and "fold" range not the same thing. Confusion ensues here for several paragraphs until the reader sees the definition of range used is the maximum/min rather than max minus min value. This definition does not come until the caption in the following table. Please define in the text to assist the reader.

We thank the reviewer for prompting us to clarify the term. We propose to define the range of variation in at the onset of the paragraph.

In the full geochemical sampling of the 8 speleothems from La Vallina Cave, Mg/Ca variation in a given stalagmite ranges between 1.7-fold and 3.5-fold, with the exception of GAE (12.2-fold range).

Fig 4 caption: for soil moisture which direction was the experiment? Do the green crosses indicate increasing or decreasing soil moisture?

We propose to clarify the figure caption by rewording:

**In a second case shown with green crosses, variable initial Ca corresponds to constant temperature, simulating decreasing soil $pCO_2$ limited by decreasing moisture at constant temperature.**

L432: there should be some earlier description of these dripwater measurements and their method, seeing as the Kost et al paper is not yet published.

As noted above, we are now able to cite the Kost et al monitoring paper.

Reference:

Silvia Frisia, Andrea Borsato, Adam Hartland, Mohammadali Faraji, Attila Demeny, Russell N. Drysdale, Christopher E. Marjo, Crystallization pathways, fabrics and the capture of climate proxies in speleothems: Examples from the tropics, Quaternary Science Reviews, Volume 297, 2022, https://doi.org/10.1016/j.quascirev.2022.107833.

Frisia, S., Borsato, A., Hartland, A., Faraji, M., Demeny, A., Drysdale, R. N., and Marjo, C. E.: Crystallization pathways, fabrics and the capture of climate proxies in speleothems: Examples from the tropics, Quaternary Science Reviews, 297, 107833, 2022.

Genty, D., Blamart, D., Ouahdi, R., Gilmour, M., Baker, A., Jouzel, J., and Van-Exter, S.: Precise dating of Dansgaard–Oeschger climate oscillations in western Europe from stalagmite data, Nature, 421, 833-837, 2003.

Genty, D., Blamart, D., Ghaleb, B., Plagnes, V., Causse, C., Bakalowicz, M., Zouari, K., Chkir, N., Hellstrom, J., and Wainer, K.: Timing and dynamics of the last deglaciation from European and North African δ13C stalagmite profiles—comparison with Chinese and South Hemisphere stalagmites, Quaternary Science Reviews, 25, 2118-2142, 2006.

Kost, O., Sliwinski, J., Gies, N., Lueder, M., and Stoll, H. M.: The influence of fluid inclusions, organics, and calcite fabric on trace element distributions in stalagmites, Frontiers in Earth Science, in review.

Li, X., Cui, X., He, D., Liao, J., and Hu, C.: Evaluation of the Heshang Cave stalagmite calcium isotope composition as a paleohydrologic proxy by comparison with the instrumental precipitation record, Scientific reports, 8, 1-7, 2018.

Sliwinski, J., Kost, O., Endres, L., Iglesias, M., Haghipour, N., González-Lemos, S., and Stoll, H.: Exploring soluble and colloidally transported trace elements in stalagmites: The strontium-yttrium connection, Geochimica et Cosmochimica Acta, 343, 64-83, 2023.

---

## Author Comment (AC2)

**Response to Reviewer Comment 1 (reviewer comment in black, response in violet and revised text in blue).**

**General comments**.

Unraveling paleoclimate signals from speleothem geochemical proxies is an important challenge to address in order to maximize applicability of such proxies. This manuscript combines TE/Ca and calcium isotopes to act as a proxy for degassing and prior calcite precipitation (PCP) and remove the effects of those two processes on stalagmite d13C records. This in turn allows one to interpret stalagmite d13C time series in terms of climate-driven variations in vegetation and soil. The authors use coeval speleothems from a single cave to produce several records to apply their models. A key part of the approach is separating the temporal trends in PCP from those due to soil/epikarst components within a stalagmite. The results show potential for d13Cinitial records to show climate shifts, and that such d13C records are difficult to use without multiple other proxies in the same speleothem. The results for multiple coeval stalagmites indicate that the δ13Cinitial approach may resolve soil/vegetation changes driven by regional climate changes.

The manuscript discusses changes in Mg/Ca and Sr/Ca (TE/Ca) in terms of two principal processes - either PCP or changing partition coefficient. The possibility of water-rock interaction (WRI) is offered as an afterthought, without adequate justification for largely discounting it as a key process to be considered. The impact of WRI on TE/Ca, and therefore on unraveling the extent of PCP, is not well addressed. It's not until page 11, line 305, that WRI is brought up as a possible factor, where it is stated that WRI is not necessary to attain the Mg/Ca range observed for the stalagmites. This misses a key point, however, that WRI – while not necessary – cannot be ruled out as an important process in karst water evolution. In this case, d13C evolution by WRI in the epikarst may occur, and change the interpretation that assumes that all above-cave processes that change water DIC d13C are due to changes in climate/vegetation/soil. There are examples in the literature of how the multiple effects of WRI and PCP can be parsed out in well understood karst systems. Quantitative approaches to this using TE/Ca and Sr isotopes can be found in Wong et al. (2011, GCA) and Sinclair et al. (2012, Chem. Geol.) There are several places where this issue is further explained in the Detailed Comments below.

We thank the reviewer for the suggestion to provide a more systematic and unified discussion of the potential variation in the trace element/Ca (TE/Ca) ratio attained as a result of bedrock dissolution prior to PCP, which was previously described in disparate sections (lines 176-180, 215-217,383-385,585-586). We propose to add a paragraph at the end of the background section 2.2.1 to clarify the role of dissolution/water rock interaction.

[revised manuscript text omitted]

The writing and referencing need work, including adding sufficient information and citations to support the points being made. The presentation and language can be clearer to explain the many new model terms used. Similarly, the figures can be improved, such as by adding error bars to age determinations and by adding more multivariate analysis for the scenarios presented. I point out some examples of this in the Detailed comments below, but my comments are by no means complete in this regard. The example

suggestions for the writing are aimed at making the scientific message clearer so that readers can get the most out of this study.

We appreciate the suggestion to include error bars to the figures showing age determinations and will implement this, together with the specific suggestions made by the reviewer later in the review.

There are 32 individual graphics within the figures. The impact of the paper does not justify this volume of material for the reader to go through. I think these and the text can be reduced by 25%.

For reducing the number of graphical elements, we propose that Table 2 could effectively move to the supplement rather than the main text. To further simplify the figures, we propose that from Figure 7, panels e, f, and g could be presented in supplement rather than main text. Likewise, from Figure 9, panel b could be presented in the supplement rather than the main text.

In summary, the approach presented is a potential useful step forward for the application of speleothem geochemical proxies. A more thorough exposition of alternative processes, such as water-rock interaction, and improved presentation is needed.

**Detailed comments by line #.**

Line 20. "the initial δ13C of dripwater". Clarify if this the dripwater once it enters the cave, or once it leaves the soil?

We thank the reviewer for prompting us to clarify this term. We propose to define the initial δ13C of dripwater as the composition prior to degassing and PCP.

1. Are the 8 stalagmites from 8 different caves? From different climate regimes? Different time periods? Why were these 8 chosen?

   In line 210 we had indicated that the reported stalagmites were all from the La Vallina Cave. We will now clarify in the introduction

   We present new analyses of PCP proxies $\delta^{44}$Ca and Mg/Ca on 9 stalagmites from a single cave system which experienced the same climate regime and represent a range of climate states from glacial through interglacial.

23-4. "Fraction of initial Ca remaining at the deposition of the stalagmite layer". How is the initial amount of Ca in fluid determined? Is this the same 'initial' as line 20 'initial d13C of dripwater'? State more clearly.

We propose to clarify as follows: .

 From $\delta^{44}$Ca and Mg/Ca, our calculation of PCP as $f_{Ca}$, fraction of initial Ca remaining in solution at the time the stalagmite layer is deposited.

24-25. State the magnitude of the uncertainty so the reader knows at the outset how well constrained (or not) this value can be using your method.

We thank the reviewer for this suggestion. We do not propose to provide this information in the abstract, as there is a range of values for different stalagmites which are illustrated in the figures.

1. "spanning THE 94 to 82 ka interval" corrected

" higher interglacial dripwater situation". This phrase does not make sense, rephrase. We thank the author for alerting us to the typo (saturation is the intended word, not situation). The revised sentence will read:

During deglaciations, the trend of greater respiration rates and higher soil $CO_2$ is captured in the calculated initial $\delta^{13}C$, despite the higher interglacial dripwater saturation favoring more extensive PCP.

2. End abstract with summary of how/where/when other speleothem studies can use this new approach.

We propose to add a sentence at the end of the abstract:

Initial $\delta^{13}C$ can be estimated for active and fossil speleothems from a range of settings, wherever there is confidence that Mg/Ca and/or $\delta^{44}Ca$ provides quantitative indication of past changes in PCP. Further study of Mg partitioning in speleothems will improve the robustness of Mg/Ca as a PCP proxy.

33-35. It will be easier for reader to unravel this building block information for the rest of the paper if this is broken up into two sentences.

We proposed the following revision of these sentences:

The $\delta^{13}C$ of $CO_2$ in the soil and karst reflects the relative contributions of isotopically light respired $CO_2$ and isotopically heavier atmospheric $CO_2$. Conditions which favor higher vegetation productivity and faster rates of heterotropic and autotrophic respiration in soils will lead to higher soil $pCO_2$ and a lower $\delta^{13}C$ of $CO_2$. In contrast, in less productive and slower respiring systems, the atmospheric $CO_2$ and its higher $\delta^{13}C$ will be more significant C sources in the soil.

40-44. This section summarizes the foundational work of Mickler et al. 2004 GCA, 2006 GSA Bulletin, and as such should cite these.

We thank the reviewer for prompting us to include these early field studies in the introduction, not only the most recent field study of Mickler (2019) cited in lines 86, 643, and 649. The revised sentence now reads:

This process has been demonstrated in lab experiments(Polag et al., 2010; Hansen et al., 2019) and in "farmed calcite" in cave environments(Mickler et al., 2004; Mickler et al., 2006) and can be modeled as a Rayleigh process.

1. Should read 'Superimposed on soil and karst processes are in-cave processes...". We infer that the reviewer is opposed to the initial comma separated clause and propose the following revision:

Superimposed on soil and karst process are in-cave processes which subsequently modify the $\delta^{13}$C of DIC and thereby speleothem $\delta^{13}$C, as coupled degassing and precipitation of CaCO₃ progressively enriches the $\delta^{13}$C of the remaining dissolved inorganic carbon.

51-3. Sr/Ca dismissed by omission here without explanation.

The rationale for focusing on Mg/Ca rather than Sr/Ca appears later in the paper and we propose to leave the introduction stating only the final employed proxies which are Mg/Ca and $\delta^{44}$Ca.

1. Eq. 1 should be checked for balance, parens, subscripts, etc. We will now write the full equation

2. Move Sade et al., 2022 citation to end of previous sentence. done

3. 'preserve', not 'conserve'. Adjusted as suggested

76-8. Be explicit in the description here. Given the quantitative, sequential, and process nature of your model – it would be clearer to explain that the Ca allows estimate of the extent of calcite precipitation, which in turn allows estimate of the degree of degassing.

Previous modeling has focused on the evolution of $\delta^{13}$C bicarbonate as a function of the remaining DIC. In contrast, here we focus on the evolution of $\delta^{13}$C bicarbonate as a function of the Ca remaining in the solution. The Ca remaining in solution allows us to estimate the extent of calcite precipitation, and from this we estimate the degree of degassing (and carbonate precipitation) the solution has undergone prior to deposition of the speleothem. We target Ca as the indicator because trace element ratios and Ca isotopic systems may preserve proxy information about the Ca remaining in solution.

83-6. More than refer to A, define it explicitly. What is a "degassing slope"? Explain here and in Table 1 definition of 'A'.

The term A describes the rate of evolution of $\delta^{13}$C with progressive degassing and CaCO₃ precipitation. We subsequently refer to term A as the degassing slope. A range of values have been proposed for A, from equilibrium precipitation and degassing in CaveCalc, to greater, kinetically-enhanced fractionation during degassing suggested by some laboratory(Hansen et al., 2019) and field (Mickler et al., 2019) studies (Figure 1b; Supplemental Figure 1). The support for each of these values and implications are discussed at length in section 5.3. We complete sensitivity test with a range of values for the parameter A.

Figure 1b. The y axis label does not indicate what phase the d13C is calculated for. We have added this to the figure axis label (it was formerly indicated only in the figure caption)

1. 'landing' better than 'impingement', which is not the proper term here. Adjusted as suggested

2. 'partition', instead of 'partitioning', change throughout manuscript. Partition coefficient is adjusted as suggested throughout.

119-20. "Such a dependence would serve to amplify the Mg/Ca due to increasing PCP". This is an important point, but it is stated cryptically. Be clearer. reworded

126-7. "Deviations from expected PCP control may reflect variation in the partitioning coefficients of Sr (or Ba)." Alternatively, such deviations may reflect other processes such as water-rock interaction (Sinclair et al. 2012, Chem. Geol.).

This section  has been fully rewritten and reorganized as presented at the onset of this response.

1. need a comma before 'which' adjusted as suggested
2. "DSr increases with increasing calcite Mg/Ca ratio", see study by Mucci and Morse, 1983, GCA 47, 217.

   We have added the following sentence and citation

   A similar increase of DSr with higher Mg/Ca, observed in laboratory growth in a seawater-like matrix, is attributed to the increased accommodation of larger Sr ions in the crystal lattice when the lattice is distorted by incorporation of Mg ions which are smaller than Ca   (Mucci and Morse, 1983).

Throughout manuscript – use convention of notation for partition coefficients with element subscripted

adjusted as suggested

Page 6, first line. See general comments above about controls of mineral-solution reactions (WRI) and host rock compositions as controls on TE/Ca values in drip waters as an important alternative mechanism. "MgCa" should read "Mg/Ca".

As detailed above, we have added a section on the dissolution process.

1. "fractionation of Ca", do you mean fractionation of Ca isotopes?, yes, we have clarified this

154-7. How is this known? Refer to a reference.  Add definition of fract factor to Table 1 and refer to that here.

We have added the citation:

Hofmann, Amy E., Ian C. Bourg, and Donald J. DePaolo. "Ion desolvation as a mechanism for kinetic isotope fractionation in aqueous systems." *Proceedings of the National Academy of Sciences* 109, no. 46 (2012): 18689-18694.

1.  "must be assumed…". Or, estimated from farmed calcite and dripwater from other sites in the same cave.

   We adjust to:

   …and must be assumed or based on information from farmed calcite in the same or similar cave settings.

174-8. To make the case for growth rate control, give the growth rates for the stalagmites from the different settings. Alternatively, if growth rate is not important, then do not provide this info for any of the stalagmites.

We remove the information on the growth rate for the Heshang stalagmite.

1. "good quantitative agreement in estimated fCa_δCa and fCa_MgCa". What estimates is this based on?

We adjust the sentence to clarify:

These two studies report good quantitative agreement in estimated fCa_$\delta$Ca and fCa_MgCa

2. "temperature and dripwater initial oversaturation may not have experienced significant variations". What is this based on? We now clarify:

   Relative to glacial-interglacial transitions, the Holocene and last interglacial experienced more stable temperatures, and potentially also more stable dripwater initial oversaturation, both factors potentially contributing to constant calcite-water $\Delta$44Ca fractionation factors.

178-80. How do you rule out variations in water/rock contact times?

We propose to expand this sentence to discuss more fully:

Heshang Cave contains dolomite host rock, and varying water residence times could potentially alter bedrock mixing fraction, altering the initial Mg/Ca of dripwater prior to degassing. However, if this process were significant, it would lead to deviation in the fCa calculated from Mg/Ca compared to that calculated from $\delta^{44}$Ca. The consistency of fCa estimates from Mg/Ca and $\delta^{44}$Ca suggest that variations in water/rock contact times did not result in appreciable variation in the bedrock mixing fraction and initial Mg/Ca from dissolution.

1. Abstract states '8' stalagmites not 9. Corrected.
2. Kost et al reference not in reference list, no date given.(see below)

215-217. This is an important point, so show the data or refer to the figure of the time series that support this statement. Without this constancy of bedrock dissolution source, the model does not work, correct?.... because it is only accounting for PCP and initial soil variability and not WRI and not host rock TE/Ca variability?

As the Kost et al reference is now published, we are able to refer to the figure which indicates the constancy of the bedrock dissolution source, Figure 12, and provide the full citation:

Kost, Oliver, Saul González-Lemos, Laura Rodríguez-Rodríguez, Jakub Sliwinski, Laura Endres, Negar Haghipour, and Heather Stoll. "Relationship of seasonal variations in drip water δ 13 C DIC, δ 18 O and trace elements with surface and physical cave conditions of La Vallina Cave, NW Spain." *Hydrology and Earth System Sciences* v 27 2023 (2022): 1-42. https://doi.org/10.5194/hess-27-2227-2023.

1. Show the age model constraints on the figures as error bars. Indicate such analytical uncertainty on figures or in caption for all figures displaying analytical results. We will modify figures and captions accordingly.

240-4. This is a run-on sentence and difficult to follow. We adjust to:

Mg/Ca measurement is much faster than $\delta^{44}Ca$, and therefore it is common to have Mg/Ca measured for every $\delta^{13}C_{init}$ measured in the stalagmite We therefore carry out examples employing Mg/Ca as the preferred measurement for deriving a continuous record of $\delta^{13}C_{init}$.

1. may also 'be' calculated: we thank the reviewer for noticing this typo, which we fix

295, and Figure 3. All but one of these data sets are comprised of only two samples, so it is of limited value to consider correlations here. Is there an overall correlation among sites, which would provide a larger data set?

Because the different drip locations may feature different bedrock mixing fractions, and also different Ca isotope fractionation factors, we cannot pool all of the data in a single figure.

1. "The 1.7 to 3.5 fold range could be fully explained by PCP".

   - Alternatively, could these values be explained by water-rock interaction? This should be addressed in the text.

   We have incorporated a substantial background on this issue in 2.2.1 and further address this in the discussion section (end of section 5.2.1, "Quantitative PCP indicators in this sample set".)

   - Is it the range of absolute values that is 1.7 – 3.5? What are the units? This is different than stating 1.7 – 3.5 fold range. There are other places in the text where units can be added for clarity. We have rewritten this section for clarity:

   In the full geochemical sampling of the 8 speleothems from Porrua Cave, Mg/Ca variation in a given stalagmite ranges between 1.7-fold and 3.5-fold, with the exception of GAE (12.2-fold range) Five of the 8 stalagmites have Mg/Ca variations of 2-fold to 3-fold (Table 2). The 1.7 to 3.5 fold range could be fully explained by PCP even with no change in DMg, since a 3.5 fold range requires the initial dripwater Ca concentration to be greater than 3.5 times the saturation Ca concentration for the cave conditions. Changes in the Mg/Ca of the initial dripwater, prior to degassing, from varying bedrock mixing fraction, or enhanced Mg/Ca ratio due to increased fluid inclusion density, are not required to attain the range of Mg/Ca in 7 of the 8 of the stalagmites.

2. 'process' should be plural.

305-6. Indeed, WR interaction is not required, but it also cannot be eliminated, and as such this process is not accounted for by the model, meaning that modeled d13C soil values may also have a component of WRI not accounted for, which would lead to inaccurate interpretations of speleothem d13C as a proxy.

The presented approach of estimating $\delta^{13}C_{init}$ using a constraint on PCP is valid. As we discuss in several sections, the potential to apply this approach to a given stalagmite depends on the robustness of the quantitative PCP indicators for the particular sample. We have recommended several criteria to assess when Mg/Ca is likely to be more robust and when it is likely to be less robust, including monitoring approaches and screening for factors such as detrital contribution to measured Mg/Ca (in section 2.2.1).

At the start of section 3.4, we discuss the advantages of $\delta^{44}Ca$ vs Mg/Ca for the quantitative estimation of PCP, including that $\delta^{44}Ca$ is much less sensitive to the bedrock mixing fraction than Mg/Ca. We also present cases where paired measurements show full agreement among the two indicators.

In section 5.2 we carefully review the coherency of Mg/Ca and $\delta^{44}Ca$ as PCP indicators, and in 5.2.2 we clearly describe the challenges and suggested approaches of using Mg/Ca for quantitative PCP estimation. Overall, we believe the manuscript presents a balanced overview of the current progress as well as uncertainties and important future lines of research to advance this approach and our accurate interpretation of $\delta^{13}C$ time series.

320, Table 2. How are the Age min and max calculated? Why are these given for Mg/Ca and not Sr/Ca?

We clarify that the table reports the age of maximum Mg/Ca and ages of minimum Mg/Ca ratios, as discussed in section 4.1. We have not added the detail for Sr/Ca to keep the table streamlined and concise, because Sr/Ca results are not employed for continuous PCP estimation.

1.  (e.g., warm climates… e.g. added
2.  Fig. 3b or Fig. 4b? Figure citation to 4b is correct

429-31. Difficult sentence to follow. Also, list ranges of values from low to high.

Because the figure employs an inverted $\delta^{13}C$ axis, we retain the current ordering of $\delta^{13}C$ values in the text.

1.  "is not well in measuredδ13C GAE". Do you mean 'is not well expressed in d13C GAE'? we have added the missing word, expressed.

Fig. 8e. There is a caption but no figure. The extraneous caption has been removed.

1.  "Because the extent of degassing and PCP can be strongly conditioned by the drip rate".. and drip chemistry adjusted as proposed

566-8. Alternatively, this could reflect water-rock interaction – see Sinclair et al. 2012.

Elsewhere in the text, we have provided the evidence for limited variation in the bedrock mixing fraction from dripwater monitoring in this cave. We now add a further citation for the partitioning effects to the sentence:

Sr/Mg ratio does not conclusively require variation in non-bedrock sources in either element but could reflect partitioning effects which for Sr are decoupled from PCP, as suggested by strong growth rate dependence of Sr partitioning in stalagmites from this cave(Sliwinski et al., 2023).

585-6. write more explicitly. This is cryptic. We now write:

In our cave system, dripwater monitoring in La Vallina system shows very constant dripwater Mg/Sr ratios seasonally despite order of magnitude changes in dripwater flow rates (Figure 12 in (Kost et al., 2023). This suggests that despite variation in water residence times, there is not widespread or significant variation in bedrock mixing fraction.

1. Mg/Ca is "a" widely…corrected

593-4. How about using monitoring of modern dripwaters instead to use Mg/Ca for quantification of PCP? We have added the sentence

589-90. "'feature significant components of wide varying solubility and Mg/Ca which give rise to strong incongruency of dissolution." This phrase after the 'and' doesn't follow. Mg/Ca of what? this is not equivalent to the first term before the 'and'. It is now revised as:

Mg/Ca is a widely measured PCP-sensitive indicator. However, it can be unreliable in host rocks which feature significant components of widely varying solubility and Mg/Ca if the bedrock mixing fraction varies over time. This can produce variation in the initial dripwater Mg/Ca attained by bedrock dissolution before evolution of dripwater by PCP. Large temporal variations in dripwater Mg/Sr ratios at a single drip during monitoring may be one sign of varying bedrock mixing fraction.

637-642. Why three paragraphs, two of which are just one sentence long? It's all the same topic. We have combined the paragraphs, as suggested.

698-699. The role of ecosystem type in the suggested approach is glossed over.

This refers to the match of interglacial values to those reported for respired endmembers in three ecosystems illustrated in Figure 6.

References cited in response

Fairchild, I. J. and Treble, P. C.: Trace elements in speleothems as recorders of environmental change, Quaternary Science Reviews, 28, 449-468, 2009.
Hansen, M., Scholz, D., Schöne, B. R., and Spötl, C.: Simulating speleothem growth in the laboratory: Determination of the stable isotope fractionation ($\delta13C$ and $\delta18O$) between H2O, DIC and CaCO3, Chemical Geology, 509, 20-44, 2019.
Kost, O., Gonzalez Lemos, S., Rodriguez-Rodriguez, L., Sliwinski, J., Endres, L., Haghipour, N., and Stoll, H. M.: Relationship of seasonal variations in drip water $\delta13CDIC$, $\delta18O$, and trace elements with surface and physical cave conditions of La Vallina cave, NW Spain, Hydrol. Earth Syst. Sci., 27, 2227-2255, https://doi.org/10.5194/hess-27-2227-2023, 2023.
Mickler, P. J., Stern, L. A., and Banner, J. L.: Large kinetic isotope effects in modern speleothems, Geological Society of America Bulletin, 118, 65-81, 2006.
Mickler, P. J., Banner, J. L., Stern, L., Asmerom, Y., Edwards, R. L., and Ito, E.: Stable isotope variations in modern tropical speleothems: evaluating equilibrium vs. kinetic isotope effects, Geochimica et Cosmochimica Acta, 68, 4381-4393, 2004.
Mickler, P. J., Carlson, P., Banner, J. L., Breecker, D. O., Stern, L., and Guilfoyle, A.: Quantifying carbon isotope disequilibrium during in-cave evolution of drip water along discreet flow paths, Geochimica et Cosmochimica Acta, 244, 182-196, 2019.
Mucci, A. and Morse, J. W.: The incorporation of Mg2+ and Sr2+ into calcite overgrowths: influences of growth rate and solution composition, Geochimica et Cosmochimica Acta, 47, 217-233, 1983.
Polag, D., Scholz, D., Mühlinghaus, C., Spötl, C., Schröder-Ritzrau, A., Segl, M., and Mangini, A.: Stable isotope fractionation in speleothems: Laboratory experiments, Chemical Geology, 279, 31-39, http://dx.doi.org/10.1016/j.chemgeo.2010.09.016, 2010.
Sinclair, D. J., Banner, J. L., Taylor, F. W., Partin, J., Jenson, J., Mylroie, J., Goddard, E., Quinn, T., Jocson, J., and Miklavič, B.: Magnesium and strontium systematics in tropical speleothems from the Western Pacific, Chemical Geology, 294, 1-17, 2012.

Sliwinski, J., Kost, O., Endres, L., Iglesias, M., Haghipour, N., González-Lemos, S., and Stoll, H.: Exploring soluble and colloidally transported trace elements in stalagmites: The strontium-yttrium connection, Geochimica et Cosmochimica Acta, 343, 64-83, 2023.

Tremaine, D. M. and Froelich, P. N.: Speleothem trace element signatures: a hydrologic geochemical study of modern cave dripwaters and farmed calcite, Geochimica et Cosmochimica Acta, 121, 522-545, 2013.